# TOWARDS FAST AND EFFECTIVE SINGLE-STEP ADVERSARIAL TRAINING

## ABSTRACT

Recently, Wong et al. (2020) showed adversarial training with single-step FGSM leads to a characteristic failure mode named *catastrophic overfitting* (CO), in which a model becomes suddenly vulnerable to multi-step attacks. Moreover, they showed adding a random perturbation prior to FGSM (RS-FGSM) seemed to be sufficient to prevent CO. However, Andriushchenko & Flammarion (2020) observed that RS-FGSM still leads to CO for larger perturbations and argue that the only contribution of the random step is to reduce the magnitude of the attacks. They suggest a regularizer (GradAlign) that avoids CO but is significantly more expensive than RS-FGSM. In this work, we methodically revisit the role of noise and clipping in single-step adversarial training. Contrary to previous intuitions, we find that *not clipping* the perturbation around the clean sample and using a *stronger noise* is highly effective in avoiding CO for large perturbation radii, despite leading to an increase in the magnitude of the attacks. Based on these observations, we propose a method called *Noise-FGSM* (N-FGSM), which attacks noise-augmented samples directly using a single-step. Empirical analyses on a large suite of experiments show that N-FGSM is able to match or surpass the performance of GradAlign while achieving a 3x speed-up.

## 1 INTRODUCTION

Deep neural networks have achieved remarkable performance on a variety of tasks (He et al., 2015; Silver et al., 2016; Devlin et al., 2019). However, it is well known that they are vulnerable to small worst-case perturbations around the input data – commonly referred to as *adversarial examples* (Szegedy et al., 2014). The existence of such adversarial examples poses a security threat to deploying models in sensitive environments (Biggio & Roli, 2018). This has motivated a large body of work towards improving the *adversarial robustness* of neural networks (Goodfellow et al., 2015; Papernot et al., 2016; Tramèr et al., 2018).

The most popular family of solutions to obtain robust neural networks is based on the concept of *adversarial training* (Goodfellow et al., 2015; Madry et al., 2018). In a nutshell, adversarial training can be posed as a min-max problem where instead of minimizing some loss over a dataset of *clean* samples, we augment the inputs with worst-case perturbations that are generated online during training. However, obtaining such perturbations is NP-hard (Weng et al., 2018) and hence, different approaches have been suggested to approximate them. They are commonly referred to as *adversarial attacks*. In their seminal work, Goodfellow et al. (2015) proposed the *Fast Gradient Sign Method* (FGSM), that generates adversarial attacks by running one step of gradient ascent on the loss function. However, while FGSM-based adversarial training provides robustness against single-step FGSM adversaries, Madry et al. (2018); Tramèr et al. (2018) showed that these models were still vulnerable to multi-step attacks, namely those allowed to perform multiple gradient ascent steps instead of a single one. Notably, Madry et al. (2018) introduced the multi-step *Projected Gradient Descent* (PGD) attack.

PGD-based attacks have now become the de facto standard for adversarial training; yet, their cost increases linearly with the number of steps. As a result, several works have focused on reducing the cost of adversarial training by approximating the worst-case perturbations with single-step attacks (Wong et al., 2020; Shafahi et al., 2019; Vivek & Babu, 2020). In particular, Wong et al. (2020) studied FGSM adversarial training and discovered that it suffers from a characteristic failure

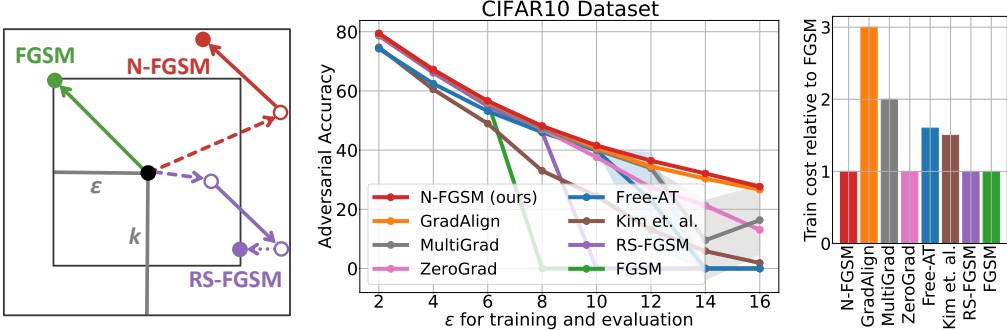

Figure 1: **Left:** Visualization of FGSM (Goodfellow et al., 2015), RS-FGSM (Wong et al., 2020) and N-FGSM (ours) attacks. While RS-FGSM is limited to noise in the $\epsilon - l_\infty$ ball, N-FGSM draws noise from an arbitrary $k - l_\infty$ ball. Moreover, N-FGSM does not clip the perturbation around the clean sample. **Middle:** Comparison of single-step methods on CIFAR-10 with PreactResNet18 over different perturbation radii ($\epsilon$ is divided by 255). Our method, N-FGSM, can match or surpass state-of-the-art results while *reducing the cost by a $3\times$ factor*. Adversarial accuracy is based on PGD-50-10 and experiments are averaged over 3 seeds. **Right**: Comparison of training costs relative to FGSM baseline based on the number of Forward-Backward passes, see Appendix K for details.

mode, in which a model suddenly becomes vulnerable to multi-step attacks despite remaining robust to single-step attacks. This phenomenon is referred to as *catastrophic overfitting*. Moreover, they argued that adding a random perturbation prior to FGSM (RS-FGSM) seemed sufficient to prevent catastrophic overfitting and yield robust models. Recently, Andriushchenko & Flammarion (2020) observed that RS-FGSM still leads to catastrophic overfitting as we increase the perturbation radii. They suggested a regularizer (GradAlign) that, on the one hand avoids catastrophic overfitting in all the settings they considered, but on the other hand requires the computation of a double derivative – which significantly increases the computational cost compared to RS-FGSM. This has motivated other works that aim at achieving the same level of robustness with a lower computational overhead (Golgooni et al., 2021; Kim et al., 2021).

In this paper, we revisit two key components that are common among previous works combining noise and FGSM (Tramèr et al., 2018; Wong et al., 2020): the role of noise, i.e. the *random step*, and the role of the *clipping step*. In Section 4.1, we study how these two components affect model robustness; our experiments suggest that adding noise with a large magnitude in the *random step* and removing the *clipping step* improves model robustness and prevents catastrophic overfitting, even against large perturbation radii. We combine these observations and propose a new method called Noise-FGSM (N-FGSM), an illustration of which is presented in Figure 1 (left). N-FGSM allows to match, or even surpass, the robust accuracy of the regularized FGSM introduced by Andriushchenko & Flammarion (2020), while *providing a 3× speed-up*.

To corroborate the effectiveness of our solution, we present an experimental survey of recently proposed single-step attacks and empirically demonstrate that N-FGSM trades-off robustness and computational cost better than other single-step approaches, evaluated over a large spectrum of perturbation radii (see Figure 1, middle and right panels), over several datasets (CIFAR-10, CIFAR-100, and SVHN) and architectures (PreActResNet18 and WideResNet28-10). We will release our code reproducing all experiments.

## 2    RELATED WORK

Since the discovery of adversarial examples, many defense mechanisms have been proposed. *Pre-processing techniques* try to modify the input image to neutralize adversarial attacks (Guo et al., 2018; Buckman et al., 2018; Song et al., 2018). *Adversarial detection* methods focus on detecting and rejecting adversarial attacks (Carlini & Wagner, 2017; Ma et al., 2018; Yang et al., 2020; Tian et al., 2021). *Certifiable defenses* provide theoretical guarantees for the lower bound performance of networks subjected to worst-case adversarial attacks, however, they incur additional costs during inference and, empirically, they yield sub-optimal performance (Cohen et al., 2019; Wong &

Kolter, 2018; Raghunathan et al., 2018; Balunovic & Vechev, 2020). *Adversarial training* methods are based on a special form of data augmentation designed to make the network robust to worst-case perturbations (Zhang et al., 2019; Athalye et al., 2018; Kurakin et al., 2017). However, computing a worst-case perturbation is an NP-hard problem that needs to be solved at every iteration. To minimize the overhead of adversarial training, Goodfellow et al. (2015) proposed FGSM which requires one additional gradient step per iteration. Tramèr et al. (2018) first proposed performing a random step before taking the adversarial step (R+FGSM), but they observed that neither method yields robust models against PGD attacks (Madry et al., 2018). Since then, augmenting the training with PGD attacks has been one of the most popular approaches for robustness, but its cost increases linearly with the number of steps, which presents a severe practical limitation.

To reduce the cost of PGD, Shafahi et al. (2019) proposed *Free Adversarial Training* (Free-AT), that exploits a single back-propagation step to both update the network parameters and compute the attack. Wong et al. (2020) explored a variation of R+FGSM, namely RS-FGSM, and showed it can yield robust networks against multi-step attacks. Andriushchenko & Flammarion (2020) found that RS-FGSM only works for limited perturbation radii and introduced GradAlign – a regularizer to linearize the loss surface. However, optimizing GradAlign *triplicates* the computational cost. This motivated a new series of works that aim at matching the performance of GradAlign without the additional computational overhead (Golgooni et al., 2021; Kim et al., 2021). Other strategies that attempted to improve FGSM included introducing dropout in every layer (Vivek & Babu, 2020) and perturbing intermediate feature maps together with the input (Park & Lee, 2021). Li et al. (2020) suggested combining RS-FGSM and PGD attacks during training, however, the proposed strategy requires a frequent monitoring of the PGD robust accuracy and, in the worst-case, is computationally equivalent to PGD training.

Gilmer et al. (2019); Fawzi et al. (2018) suggested a strong link between robustness to adversarial attacks and to random noise. Motivated by this, we revisit the idea of combining noise and FGSM and propose N-FGSM. Our method is closely related to RS-FGSM, however, we find that using a larger amount of noise *and* removing the constraint that attacks must lie in the $\epsilon - l_\infty$ ball is key to obtaining robust models. We note that Kang & Moosavi-Dezfooli (2021) concurrently studied RS-FGSM without clipping, however, as opposed to our work, they did not investigate and provide insights on the impact of noise, and the learned models were not robust against large perturbations.

## 3 PRELIMINARIES ON SINGLE-STEP ADVERSARIAL TRAINING

Given a classifier $f_\theta : \mathcal{X} \to \mathcal{Y}$ parameterized by $\theta$ and a perturbation set $\mathcal{S}$, the classifier $f_\theta$ is said to be *robust* at $x \in \mathcal{X}$ under $\mathcal{S}$ if the following holds for all $\delta \in \mathcal{S}$: $f_\theta(x + \delta) = f_\theta(x)$. One of the most popular definitions for $\mathcal{S}$ is the $\epsilon - \ell_\infty$ ball, i.e. $\mathcal{S} = \{\delta : \|\delta\|_\infty \leq \epsilon\}$. This is known as the $l_\infty$ threat model and is the setting we adopt throughout this work.

To train networks that are robust against $\ell_\infty$ threat models, adversarial training modifies the classical training procedure of minimizing a loss function over a dataset $\mathcal{D} = \{(x_i, y_i)\}_{i=1:N}$ of images $x_i \in \mathcal{X}$ and labels $y_i \in \mathcal{Y}$. In particular, adversarial training instead minimizes the worst-case loss over the perturbation set $\mathcal{S}$, i.e. trains on adversarially perturbed samples $\{(x_i + \delta_i, y_i)\}_{i=1:N}$. When using the $l_\infty$ threat model, we can formalize adversarial training as solving the following problem:

$$\min_\theta \sum_{i=1}^N \max_\delta \mathcal{L}(f_\theta(x_i + \delta), y_i), \quad \text{subject to } \|\delta\|_\infty \leq \epsilon, \tag{1}$$

where $\mathcal{L}$ is typically the cross-entropy loss for image-classification models. Due to the difficulty of finding the exact inner maximizer, the most common procedure for adversarial training is to approximate the worst-case perturbation through several PGD steps (Madry et al., 2018). While this has been shown to yield robust models, it comes at a cost of a linear increase in the computational overhead with the number of PGD steps. As a result, several works have focused on reducing the cost of adversarial training by approximating the inner maximization with single-step attacks.

If we assume that the loss function is locally linear with respect to changes in the input, then the inner maximization of Equation (1) enjoys a closed form solution. Goodfellow et al. (2015) leveraged this result to propose the FGSM method, which takes one step in the direction of the sign of the gradient. Tramèr et al. (2018) proposed adding a random initialization prior to FGSM. However, both

methods were later shown to be vulnerable against multi-step attacks, such as PGD (Madry et al., 2018). Contrary to prior intuition, recent work from Wong et al. (2020) observed that combining a random step with FGSM can actually lead to a promising robustness performance. In particular, we note that most recent single-step methods approximate the worst-case perturbation that results from solving the inner maximization problem in Equation (1) with the following general form:

$$\delta = \psi\Big(\eta + \alpha \cdot \text{sign}\big(\nabla_{x_i}\mathcal{L}(f_\theta(x_i + \eta), y_i)\big)\Big), \quad \text{where} \ \ \eta \sim \Omega \tag{2}$$

and $\Omega$ is the distribution from which we draw noise perturbations. For example, when $\psi$ is the projection operator onto the $\ell_\infty$ ball and $\Omega$ is the uniform distribution in $[-\epsilon, \epsilon]$, this recovers RS-FGSM with the following update:

$$\delta_{\text{RS-FGSM}} = \text{Proj}_{\|\delta\|_\infty \leq \epsilon}\Big(\eta + \alpha \cdot \text{sign}\big(\nabla_{x_i}\mathcal{L}(f(x_i + \eta), y_i)\big)\Big), \quad \text{where} \ \ \eta \sim \mathcal{U}[-\epsilon, \epsilon]^d. \tag{3}$$

On the other hand, with a different noise setting where $\Omega = (\epsilon - \alpha) \cdot \text{sign}\left(\mathcal{N}(\mathbf{0}, \mathbf{I})\right)$ and by choosing the step size $\alpha$ to be in $[0, \epsilon]$ we recover R+FGSM by Tramèr et al. (2018) that initially explored the idea of combining noise with FGSM but reported no improvement over adversarial training with FGSM. If we consider $\Omega$ to be deterministically 0 and $\psi$ to be the identity map, we recover the FGSM. Finally, if we adjust the choice of the loss function $\mathcal{L}$ to include a gradient alignment regularizer, this recovers the GradAlign algorithm by Andriushchenko & Flammarion (2020).

## 4 Noise and FGSM

A common practice when performing adversarial training is to restrict the perturbations used during training to the same $\epsilon - \ell_\infty$ ball that will be considered at test time. The rationale behind it is that increasing the magnitude of perturbations could "unnecessarily" decrease the clean accuracy, since perturbations outside the ball will not be evaluated at test time. For instance, R+FGSM combines the noise step, with magnitude $(\epsilon - \alpha)$, and the FGSM step, with magnitude $\alpha$, in a convex combination manner, thereof, restricting the perturbation to $\epsilon$. On the other hand, Wong et al. (2020) apply a clipping operation after the FGSM step to the $\epsilon - \ell_\infty$ ball. In the following, we experimentally challenge this common practice and explore the two key components in previous single-step methods that limit the magnitude of the perturbations. In particular, we explore **(i)** the role of the clipping operation, i.e. $\psi$ as a projection to $\ell_\infty$ ball; and **(ii)** the source and magnitude of noise for the random step, i.e. $\Omega$. We thoroughly revisit the role of both components on the robustness attained in single-step methods. Throughout this work, unless stated otherwise, we consider noise perturbations $\eta$ sampled from a symmetric Uniform distribution, i.e. $\Omega = \mathcal{U}[-k, k]^d$, where $d$ is the dimension of $\mathcal{X}$ and we refer to $k$ as the "noise magnitude".

### 4.1 The Role of Noise and Clipping on Robustness in Single-Step Methods

**Clipping Reduces the Effectiveness of Single-Step Perturbations.** To study the impact clipping has on model robustness, we consider the training of PreActResNet18 (He et al., 2016) on CIFAR-10 (Krizhevsky & Hinton, 2009) with an instance of Equation (2) as a single-step adversarial training. In particular, we consider the case where $\psi$ is a projection to the $\ell_\infty$ ball of size $\epsilon_{\text{clip}} \in \{\epsilon, 2\epsilon, 3\epsilon, \infty\}$, where $\infty$ denotes that $\psi$ is an identity function, i.e. no clipping is performed. Moreover, we consider noise sampled from a symmetric uniform distribution where $k \in \{\epsilon, 2\epsilon, 3\epsilon, 4\epsilon\}$. We report in Figure 2 the robust accuracy using PGD-50-10 (i.e. PGD attack with 50 iterations and 10 restarts) with $\epsilon = 8/255$. We observe in Figure 2 (left), that for all choices of noise magnitude $k$, as we expand the clipping $\ell_\infty$ ball, i.e. we increase $\epsilon_{\text{clip}}$, the adversarial robustness improves. We believe that this is due to the fact that more aggressive clipping, i.e. smaller $\epsilon_{\text{clip}}$, reduces the strength of the computed single-step perturbations during training. To support this intuition, we report in Figure 2 (middle) the distribution of the loss measured at perturbed points prior to applying the clipping step and after applying the clipping step, with $\epsilon_{\text{clip}} = \epsilon$. Moreover, the negative impact of clipping during training is more prominent as we increase the noise magnitude $k$. This is to be expected since, for a fixed $\alpha$, increasing the noise magnitude $k$ will lead to a prevalence of the noise component over the sign gradient direction in Equation (2).

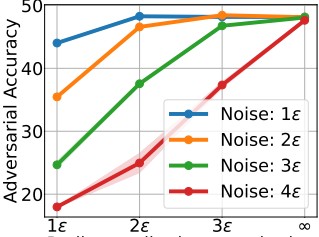 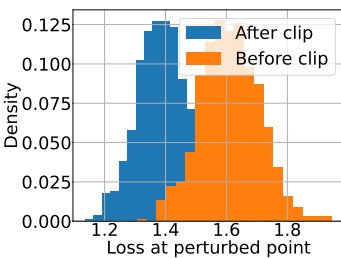 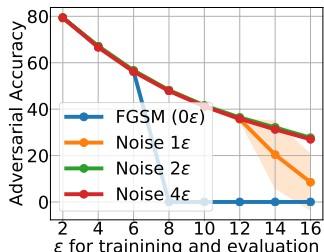

Figure 2: **Left:** N-FGSM + Clipping to different radii ($\infty$ means no clipping). As we constraint perturbations by reducing the clipping radius, adversarial accuracy drops. This effect is stronger as we increase the noise magnitude. Thus, clipping seems to have a negative impact on robustness. **Middle:** Histogram of the loss value for perturbations before and after clipping to the $\epsilon - \ell_\infty$ ball. There is a clear shift in the distributions, which indicates that clipping reduces the adversarial effect of perturbations. **Right**: N-FGSM when varying the noise magnitude $k$ ($\epsilon$ is divided by 255). Increasing the amount of noise is key to avoiding catastrophic overfitting. For (left) and (right) plots, adversarial accuracy is based on PGD-50-10 and experiments are averaged over 3 seeds.

Thus, overall, we observe that clipping during training has negative impact on the robust accuracy. Despite unclipped perturbations may lie outside the $\epsilon - \ell_\infty$ ball, we could not observe any significant drop in clean accuracy. In fact, we obtain comparable clean accuracy to GradAlign as later reported in Section 5.5. Further analyses can be found in Appendix C.

**The role of noise in single-step adversarial training.** Neither Tramèr et al. (2018) nor Wong et al. (2020) explored settings with an increased noise magnitude. Moreover, Andriushchenko & Flammarion (2020) argue that noise in RS-FGSM is not important per se, claiming that its main purpose is only reducing the $\ell_2$ norm of the final perturbation $\delta_{\text{RS-FGSM}}$ so that the loss is still in the linear regime. However, we empirically find that increasing the noise magnitude is key to avoiding catastrophic overfitting – see Figure 2 (right).

Investigating further, similarly to Kim et al. (2021), we plot the loss surface at the end of training (see Figure 12 in Appendix J) and find that, as observed by Kim et al. (2021), the loss surface of models trained via FGSM or RS-FGSM appears distorted at the end of training, i.e. the loss increases sharply along the FGSM direction, but then it rapidly decreases back to the same loss values associated with the clean sample. This is consistent with the gradient obfuscation observed by Tramèr et al. (2018). However, when training without clipping with an increased noise magnitude, we observe a non-distorted loss surface, i.e. the loss gradually increases along the FGSM direction. Interestingly, we observe a similar effect when training with the GradAlign regularizer. Thus, combining noise with FGSM *seems* to have a regularizing effect that encourages the loss surface to be locally linear in a similar spirit to GradAlign. Note this is based on the empirical analyses and visual inspections that we performed. We do not provide theoretical justification behind this.

Despite the clear benefits of increasing the noise magnitude, we do observe a slight but consistent decrease in robustness as we keep on increasing the noise, which we hypothesize is due to over-regularization. We find $k = 2\epsilon$ to be the sweet spot in most of our experiments; we do not exclude that a more extensive hyperparameter tuning procedure may lead to improved results.

## 4.2 OUR APPROACH

In the previous section, we provided experimental analyses to show that increasing the noise magnitude and not clipping prevents catastrophic overfitting and improves robustness significantly. Based on these observations, we propose the following, simple and efficient, single-step method for adversarial training that we denote as Noise-FGSM (N-FGSM):

$$\delta_{\text{N-FGSM}} = \eta + \alpha \cdot \text{sign}\big(\nabla_{x_i}\mathcal{L}(f_\theta(x_i + \eta), y_i)\big), \quad \eta \sim \mathcal{U}[-k, k]^d. \tag{4}$$

We detail our full adversarial training procedure in Algorithm 1.

**Theoretical insights** We now theoretically analyse N-FGSM to understand the role of noise in single-step approaches. According to Andriushchenko & Flammarion (2020), the main contribution

---

**Algorithm 1** N-FGSM adversarial training

---

1: **Inputs:** epochs $T$, batches $M$, radius $\epsilon$, step-size $\alpha$ (default:$\epsilon$), noise magnitude $k$ (default:$2\epsilon$).
2: **for** $t = 1, \ldots, T$ **do**
3:      **for** $i = 1, \ldots, M$ **do**
4:          *// Perform N-FGSM adversarial attack*
5:          $\eta \sim \text{Uniform}[-k, k]^d$
6:          $\delta = \eta + \alpha \cdot \text{sign}\big(\nabla_{x_i}\mathcal{L}(f_\theta(x_i + \eta), y_i)\big)$
7:          $\nabla_\theta = \nabla_\theta \mathcal{L}(f_\theta(x_i + \delta), y_i)$
8:          $\theta = \text{optimizer}(\theta, \nabla_\theta)$ *// standard weight update, (e.g. SGD)*
9:      **end for**
10: **end for**

---

of noise in single-step adversarial training is to reduce the effective $\ell_2$ norm of the perturbation $\delta$. Since N-FGSM does not involve clipping, we find the expected norm squared of N-FGSM perturbations to be larger than RS-FGSM perturbations. In fact, it is even larger than the FGSM perturbations which always lie on the $\epsilon - \ell_\infty$ ball. These observations are formalized in Theorem 1.

**Theorem 1.** *Let $\delta_{\text{N-FGSM}}$ be our proposed single-step method defined by Equation* (4)*, $\delta_{\text{FGSM}}$ be the FGSM method (Goodfellow et al., 2015) and $\delta_{\text{RS-FGSM}}$ be the RS-FGSM method (Wong et al., 2020). Then, with default hyperparameter values and for any $\epsilon > 0$, we have that*

$$\mathbb{E}_\eta \left[ \|\delta_{\text{N-FGSM}}\|_2^2 \right] > \mathbb{E}_\eta \left[ \|\delta_{\text{FGSM}}\|_2^2 \right] > \mathbb{E}_\eta \left[ \|\delta_{\text{RS-FGSM}}\|_2^2 \right].$$

We present the *Proof* in Appendix F where we also compute an empirical estimation of the expected $\ell_2$ norm of different methods via Monte Carlo sampling and find them to align with Theorem 1. Differently from the hypothesis of Andriushchenko & Flammarion (2020), we find that despite that N-FGSM perturbations have larger $\ell_2$ norms, they yield robust models even under larger $\epsilon$ radii, for which RS-FGSM or FGSM fail to catastrophic overfitting. We believe that these contradictory observations can lead to a better understanding on the role of noise in adversarial training and catastrophic overfitting in future work.

## 5 EXPERIMENTS AND ANALYSES

We compare N-FGSM against several adversarial training methods, considering a broad range of $\epsilon - l_\infty$ radii. Following Wong et al. (2020); Andriushchenko & Flammarion (2020), we measure adversarial robustness on CIFAR-10/100 (Krizhevsky & Hinton, 2009) and SVHN (Netzer et al., 2011) datasets with PGD-50-10 attack – PGD (Madry et al., 2018) with 50 iterations and 10 restarts.

### 5.1 COMPARISON TO OTHER SINGLE-STEP METHODS

We start by comparing N-FGSM against other single-step methods. Note that not all single-step methods are equally expensive, since they may involve more or less computationally demanding operations. For instance, GradAlign (Andriushchenko & Flammarion, 2020) relies on a regularizer that is considerably expensive, while, MultiGrad (Golgooni et al., 2021) requires evaluating the input gradients on multiple random points. For a comparison of training cost over different single-step methods, we refer to Figure 1 (right). Following the standard practice (Wong et al., 2020; Andriushchenko & Flammarion, 2020), we use a PreactResNet18 architecture (He et al., 2016).

We use RS-FGSM and Free-AT with the settings recommended by Wong et al. (2020). We apply GradAlign following the hyperparameters reported in the official repository [1]. Golgooni et al. (2021) recommend applying MultiGrad with $n = 3$ random samples, but do not provide a default hyperparameter setting for what concerns the ZeroGrad variant. Also Kim et al. (2021) do not recommend a set of hyperparameters; for a fair comparison, we ablate them and select the ones that provide the highest adversarial accuracy (for every combination of $\epsilon$ and dataset). We train on CIFAR-10/100 for 30 epochs and on SVHN for 15 epochs with a cyclic learning rate. Only for Free-AT, we use 96 and 48 epochs for CIFAR-10/100 and SVHN, respectively, to obtain comparable results following

---

[1]https://github.com/tml-epfl/understanding-fast-adv-training/

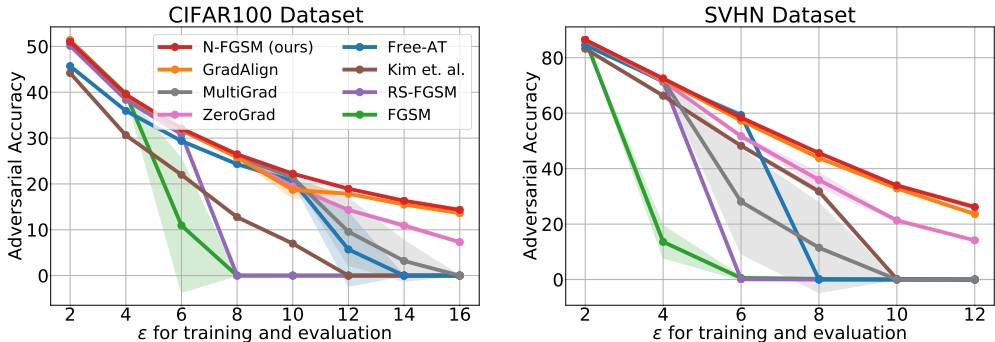

Figure 3: Comparison of single-step methods on CIFAR-100 (left) and SVHN (right) with Preact-ResNet18 over different perturbation radius ($\epsilon$ is divided by 255). Our method, N-FGSM, can match or surpass state-of-the-art results while *reducing the cost by a $3\times$ factor*. Adversarial accuracy is based on PGD-50-10 and experiments are averaged over 3 seeds. Legend is shared among plots.

Wong et al. (2020) and Andriushchenko & Flammarion (2020). CIFAR-10 results are presented in Figure 1 (middle), whereas CIFAR-100 and SVHN results are reported in Figure 3.

As previously observed, FGSM and RS-FGSM both suffer from catastrophic overfitting on larger $\epsilon$ attacks. In contrast, N-FGSM prevents catastrophic overfitting and enjoys robustness properties comparable or superior to GradAlign for all $\epsilon$, *while being 3 times faster*. With appropriate hyperparameters, ZeroGrad is able to avoid catastrophic overfitting but obtains sub-optimal robustness – especially for large perturbations. Neither MultiGrad nor the method proposed by Kim et al. (2021) can avoid catastrophic overfitting in all settings. We also observe that Free-AT cannot overcome catastrophic overfitting as also observed by Andriushchenko & Flammarion (2020).

## 5.2 RANDOMIZED ALPHA

Kim et al. (2021) evaluate intermediate points along the RS-FGSM direction in order to pick the "optimal" perturbation size. However, we find that increasing the number of intermediate evaluated points does not necessarily lead to increased adversarial accuracy. Moreover, for large perturbations we could not prevent catastrophic overfitting even with twice the number of evaluations tested by Kim et al. (2021). This motivates us to test a very simple baseline where instead of evaluating intermediate steps, the RS-FGSM perturbation size is randomly selected as: $\delta = t \cdot \delta_{\text{RS-FGSM}}$ where $t \sim \mathcal{U}[0,1]^d$. Interestingly, as reported in Figure 4 (left), we find that this very simple baseline, dubbed *RandAlpha*, is able to avoid catastrophic overfitting for all values of $\epsilon$ and outperforms Kim et al. (2021) on CIFAR-10. This is aligned with our main finding that combining noise with adversarial attacks is indeed a powerful tool that should be explored more thoroughly before developing more expensive solutions. We reach the same conclusions for CIFAR-100 and SVHN in Appendix H.

## 5.3 HYPERPARAMETER SELECTION

While FGSM relies on a fixed step-size (equal to the maximum radius of perturbation to be used at test time, i.e. $\alpha = \epsilon$), Wong et al. (2020) explored different values for $\alpha$ during the development of RS-FGSM finding that an increase of the step-size improves the adversarial accuracy – up to a magnitude before catastrophic overfitting occurs. We also ablate the value of $\alpha$ for N-FGSM in Figure 4 (middle). We find that by increasing the noise magnitude, N-FGSM can use larger $\alpha$ values than RS-FGSM, without suffering from catastrophic overfitting. As observed by Wong et al. (2020), this in turn leads to an increase in the adversarial accuracy at the expense of a decrease in the clean accuracy. In light of this trade-off and following FGSM, we also use $\alpha = \epsilon$ for N-FGSM.

Regarding N-FGSM noise hyperparameter, we find $k = 2\epsilon$ works in all but one SVHN experiment ($\epsilon = 12$, in which we set $k = 3\epsilon$), reducing the need for expensive hyperparameter tuning. In comparison, GradAlign regularizer hyperparameter or ZeroGrad quantile value need to be defined for every radius with a noticeable shift between CIFAR-10 and SVHN hyperparameters, suggesting they may require additional tuning when applied to novel datasets.

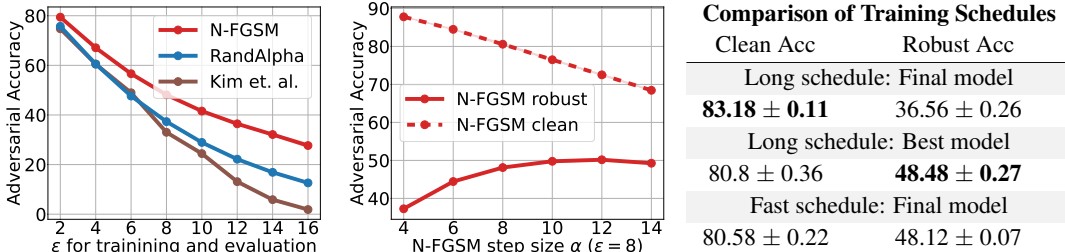

Figure 4: **Left:** Comparison of Kim et al. (2021) with *RandAlpha*, our baseline where we multiply the RS-FGSM perturbation by a value drawn uniformly in [0, 1], on CIFAR-10 and PreActResNet18 ($\epsilon$ is divided by 255). RandAlpha does not incur the extra cost of evaluating intermediate steps and does not require hyperparameter tuning. **Middle:** Ablation of step size $\alpha$ in N-FGSM for $\epsilon = 8$. As we increase the magnitude of the FGSM perturbation we observe an increase in robustness coupled with a drop on the clean accuracy. **Right:** Comparison of the "fast" training schedule from Wong et al. (2020) and "long" training schedule described in Rice et al. (2020). N-FGSM shows robust oberfitting but not catastrophic overfitting with the long schedule. Adversarial accuracy is based on PGD-50-10 and experiments are averaged over 3 seeds.

## 5.4 LONG VS FAST TRAINING SCHEDULE

Throughout our experiments, we used the RS-FGSM training setting introduced in Wong et al. (2020). However, Rice et al. (2020) suggest that a longer training schedule coupled with early stopping may lead to a boost in performance. Kim et al. (2021) and Li et al. (2020) report that longer training schedules increase the chances of catastrophic overfitting for RS-FGSM and that this limits its performance. We test the longer training schedule with N-FGSM and find that it presents *robust overfitting*, i.e. the adversarial accuracy on the training set keeps increasing but it decreases when evaluated on the test set as described in Rice et al. (2020). However, it does not suffer from catastrophic overfitting. In Figure 4 (right), we show the results for $\epsilon = 8$. Although we do observe a slight increase in performance when using the long training schedule, we find the performance remarkably competitive when considering the fast one which seems to avoid robust overfitting (when relying on early stopping, the best models are usually found at the end). Thus, it might be preferable to consider the fast training schedule if computational cost is an important factor. In Appendix E, we report a robust accuracy of $47.86 \pm 0.1$ for GradAlign with the longer schedule compared to $48.48 \pm 0.27$ for N-FGSM. Note that, in order to prevent GradAlign from suffering from catastrophic overfitting for $\epsilon = 8$, we had to increase the regularizer hyperparameter (compared to the fast schedule), while N-FGSM is able to prevent catastrophic overfitting with the same settings.

## 5.5 COMPARISON TO MULTI-STEP ATTACKS

In Section 5.1, we compared the performance of single-step methods and observed that N-FGSM is able to match or surpass the state-of-the-art method, i.e. GradAlign, while reducing the computational cost by a factor of $3\times$. In this section, we compare the performance of N-FGSM with multi-step attacks. We use PGD-2 with $\alpha = \epsilon/2$ and PGD-10 with $\alpha = 2$ following Wong et al. (2020) and keep the same training settings as described in Section 5.1 (note that PGD-x denotes PGD attack with x iterations and no restarts).

In Figure 5, we observe that PGD-2 also presents catastrophic overfitting when we increase the perturbation radius $\epsilon$, which is consistent with results reported by Andriushchenko & Flammarion (2020). On the other hand, despite all methods achieving comparable clean accuracy, there is a gap on adversarial accuracy between PGD-10 and single-step methods which grows with the perturbation size. This can be partially expected since the search space grows exponentially with $\epsilon$ and PGD can explore it more thoroughly as we increase the number of iterations. Nevertheless, *computing a PGD-10 attack is $10\times$ more expensive than computing an N-FGSM one*. An important direction for future work should be addressing this gap and analyse, both theoretically and empirically, whether single-step methods can actually match the performance of their multi-step counterparts.

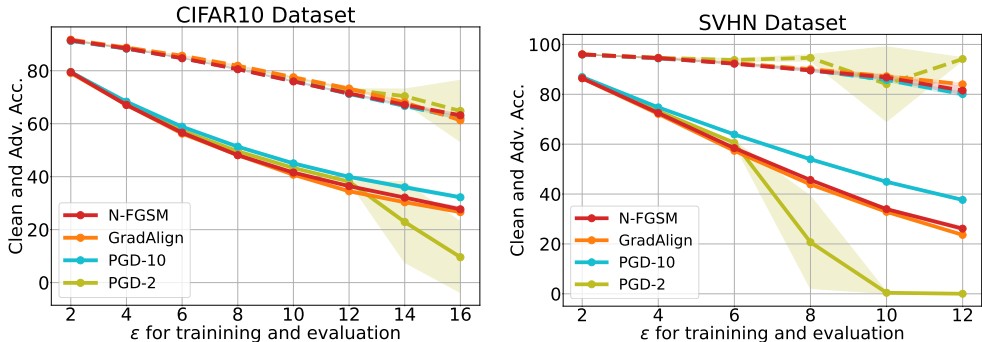

Figure 5: Comparison of N-FGSM and GradAlign with multi-step methods on CIFAR-10 (Left) and SVHN (Right) with PreactResNet18 over different perturbation radii ($\epsilon$ is divided by 255). Despite all methods achieving comparable clean accuracy (dashed lines), there is a gap in robust accuracy between PGD-10 and single-step methods. However, note that PGD-10 is $10\times$ more expensive than N-FGSM. Adversarial accuracy is based on PGD-50-10 and experiments are averaged over 3 seeds.

### 5.6 EXPERIMENTS WITH WIDERESNET28-10 ARCHITECTURE

We also compare the performance of all methods on WideResNet28-10 (Zagoruyko & Komodakis, 2016) architecture in Figure 7 and Figure 8 (Appendix B). As in the experiments with the PreAct-ResNet18 architecture, N-FGSM obtains state-of-the-art PGD-50-10 accuracy among single-step methods. Nevertheless, as a general trend, we observe that catastrophic overfitting seems to be more difficult to prevent when using WideResNet. For instance, FGSM is able to consistently yield robust models up to $\epsilon = 6$ for PreActResNet18 on CIFAR-10, however, for some runs the same radius can lead to catastrophic overfitting for WideResNet models. We hypothesize this is because it is more over-parametrized –WideResNet28-10 has 36.5 M parameters, whereas PreActResNet18 has 11.2M. Regarding GradAlign, we had to increase the regularizer hyperparameter (compared to the settings for PreActResNet18) in order to prevent catastrophic overfitting on CIFAR-100. Note that, to our surprise, *we could not find a competitive hyperparameter setting for GradAlign on the SVHN dataset* for $\epsilon \geq 6$. We tried both increasing the regularizer hyperparameter and decreasing the step size $\alpha$, but some or all runs led to models close to a constant classifier for each setting. We do not claim that GradAlign will not work, but finding a good configuration might require further tuning.

On the other hand, as observed earlier, the default configuration for N-FGSM ($\alpha = \epsilon$, $k = 2\epsilon$) works well in all settings except for $\epsilon = 16$ on CIFAR-10 and $\epsilon = 10$, 12 on SVHN. For CIFAR-10, we increase the noise magnitude to $k = 4\epsilon$. For SVHN, we find that decreasing $\alpha$ as we tried for GradAlign works better than increasing the noise. However, in both cases N-FGSM can obtain more than a trivial adversarial accuracy. Results are presented in Appendix B.

## 6 DISCUSSION

In this work, we explore the role of noise and clipping in single-step adversarial training. Contrary to previous intuitions, we show that increasing the noise magnitude and removing the $\epsilon - \ell_\infty$ constraint allows improving adversarial robustness, while maintaining a competitive clean accuracy. These findings led us to propose N-FGSM, a simple and effective approach that can match or surpass the performance of GradAlign (Andriushchenko & Flammarion, 2020), while achieving a $3\times$ speed-up.

We perform an extensive comparison with other relevant single-step methods, observing that all of them achieve sub-optimal performance and most of them are not able to avoid catastrophic overfitting for larger $\epsilon$ attacks. Moreover, we also analyze recent single-step methods and – inspired by Kim et al. (2021) – observe that uniformly choosing a step-size in $[0, \alpha]$ avoids catastrophic overfitting for RS-FGSM, which reinforces our intuition that random noise is a powerful tool towards learning robust models. However, despite impressive improvements of single-step adversarial training methods, there is still a gap between single-step and multi-step methods such as PGD-10 as we increase the $\epsilon$ radius. Therefore, future work should put an emphasis on formally understanding the limitations of single-step adversarial training and explore how, if possible, this gap can be reduced.

**Ethics Statement.** The existence of adversarial examples poses a potential threat to the deployment of deep learning systems into the real world. Therefore, finding fast and effective methods to train models robust against this threat is of utmost importance. On the other hand, adversarial training methods are based on augmenting samples with adversarial perturbations, thus, they are partially based on building methods to attack neural networks. Research on adversarial attacks is naturally a sensitive path, since it can potentially be exploited for unethical purposes. However, the scope of our work is not designing stronger attacks; rather, the methods we propose are designed ad hoc to improve the adversarial robustness of learning systems, not to break other models' defenses. Therefore, although we are aware that we are carrying out research within a sensitive topic, we are not particularly concerned that the research presented in this paper can lead to harmful applications – on the contrary, we believe that it can help deploying safer machine learning applications.

**Reproducibility Statement.** In this paper, we compare against several adversarial training methods. The general training and evaluation settings used are described at the beginning of Section 5. Moreover, different hyperparameters have been used for each method depending on the dataset and network. These are detailed in the manuscript, where corresponding results are discussed. For all the reported experiments, we employ different random seeds to ensure replicability of our results, and report performances indicating the standard deviation values. To facilitate further research on this topic, as stated in Section 1, we will release the code to reproduce all experiments. Regarding our theoretical results, we deferred the proofs to Appendix F.

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

## A  ADDITIONAL PLOTS FOR PREACTRESNET18 EXPERIMENTS

In the main paper we compare N-FGSM with other single-step methods and multi-step methods separately and remove clean accuracies for better visualization. In this section we present the curves for all methods with both the clean and robust accuracy. The tendency in the three datasets is for N-FGSM PGD-50-10 accuracy to be slightly above that of GradAlign, while the opposite happens to the clean accuracy. We also observe that clean accuracy becomes significantly more noisy when catastrophic overfitting happens. Exact numbers for all the curves are in Appendix L.

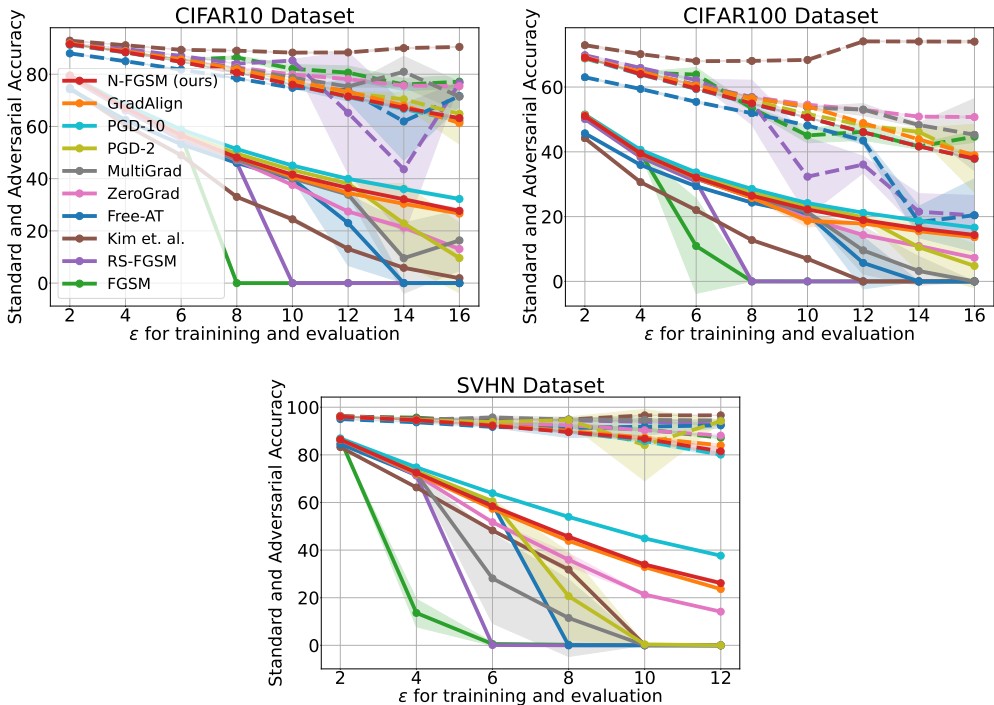

Figure 6: Comparison of all methods on CIFAR-10, CIFAR-100 and SVHN with PreactResNet18 over different perturbation radius ($\epsilon$ is divided by 255). We plot both the robust (solid line) and the clean (dashed line) accuracy for each method. Our method, N-FGSM, is able to match or surpass the state-of-the-art single-step method GradAlign while *reducing the cost by a* $3\times$ *factor*. Adversarial accuracy is based on PGD-50-10 and experiments are averaged over 3 seeds. Legend is shared among all plots.

## B  EXPERIMENTS WITH WIDERESNET28-10 ARCHITECTURE

In this section we present the plots of our experiments with WideResNet28-10. We report the results in two figures. In Figure 7 we compare all single-step methods and we do not plot the clean accuracy for better visualization. In Figure 8 we plot all methods, including multi-step methods, and report the clean accuracy as well with dashed lines. Since we observed that our baseline, RandAlpha, outperformed Kim et al. (2021) in all settings for PreActResNet18, we only report RandAlpha for WideResNet. As mentioned in the main paper, we observe that catastrophic overfitting seems to be more difficult to prevent for WideResNet. In particular, for GradAlign we observed the regularizer hyperparameter settings proposed by Andriushchenko & Flammarion (2020) for CIFAR-10 (searched for a PreActResNet18) worked well. However, those parameters led to Catastrophic Overfitting for $6 \le \epsilon \le 12$ in CIFAR-100. Since $\epsilon = 14, 16$ did not show Catastrophic Overfitting, we increased the GradAlign regularizer hyperparameter $\lambda$ for CIFAR-100 so that each $6 \le \epsilon \le 12$ would have the default value corresponding to $\epsilon + 2$, for instance, $\lambda$ for $\epsilon = 6$ would be the default $\lambda$ in Andriushchenko & Flammarion (2020) for $\epsilon = 8$.

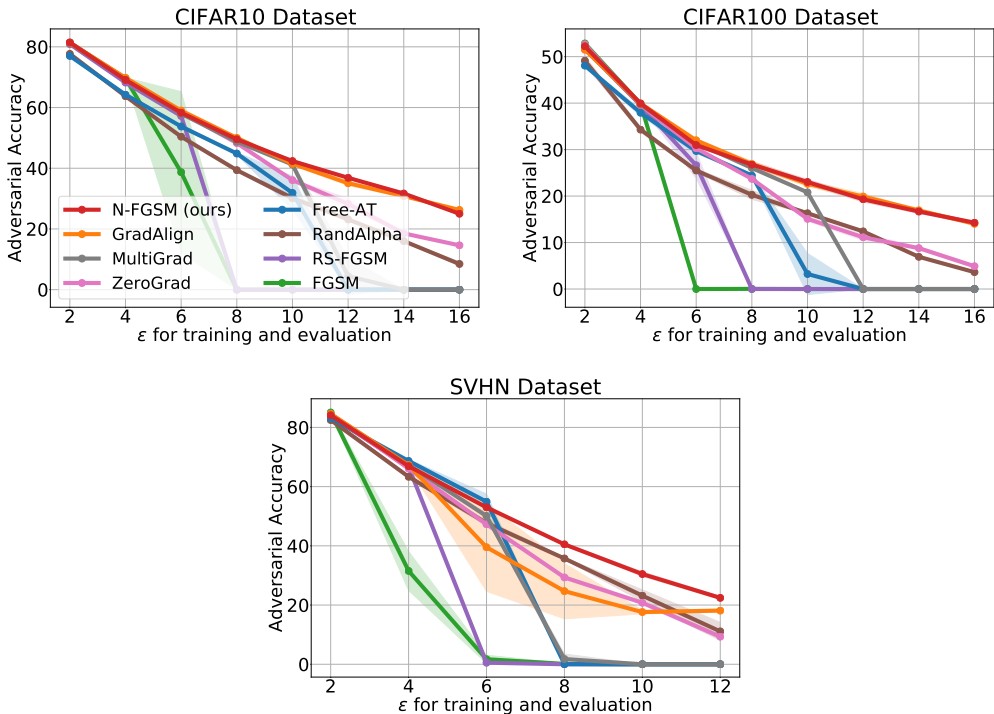

Figure 7: Comparison of single-step methods on CIFAR-10, CIFAR-100 and SVHN with WideResNet28-10 over different perturbation radius ($\epsilon$ is divided by 255). Our method, N-FGSM, is able to match or surpass the state-of-the-art single-step method GradAlign while *reducing the cost by a $3\times$ factor*. Moreover, we could not find any competitive hyperparameter setting for GradAlign for $\epsilon \geq 6$ in SVHN dataset. Adversarial accuracy is based on PGD-50-10 and experiments are averaged over 3 seeds. Legend is shared among all plots.

For SVHN we observed that the default values for $\lambda$ led to models close to a constant classifier for $\epsilon \geq 6$. We tried to increase the lambda for those $\epsilon$ values to $1.25\lambda$ but observed the same result. Since the model did not show typical catastrophic overfitting but rather it seemed as it was underfitting, we tried to reduce the step-size to $\alpha = 0.75\epsilon$ and also both decreasing $\alpha$ and increasing $\lambda$. When reducing the step size we obtain accuracies above those of a constant classifier for some radii, however, some or all seeds converge to a constant classifier for each setting, hence the large standard deviations. For N-FGSM, the default configuration of N-FGSM ($\alpha = \epsilon$, $k = 2\epsilon$) works well in all settings except for $\epsilon = 16$ on CIFAR-10 and $\epsilon = 10, 12$ on SVHN. For CIFAR-10, we increase the noise magnitude to $k = 4\epsilon$. For SVHN we find that decreasing $\alpha$ as we tried for GradAlign works better than increasing the noise. We use $\alpha = 8$ for both $\epsilon$ radii. Exact numbers for all the curves are in Appendix L

## C FURTHER ABLATION OF NOISE AND STEP SIZE IN N-FGSM

In Section 4.1, we observed that both removing clipping and increasing noise were necessary to avoid catastrophic overfitting. However, when doing that, we increase the magnitude of the perturbations. In this section, we study more closely the interplay between the step-size $\alpha$ and the noise level $k$ to ensure that it is indeed the increase in noise level, rather than merely increasing the magnitude of perturbations, what helps stabilize adversarial training and avoid catastrophic overfitting.

We fix $\epsilon_{\text{test}} = 8/255$ for evaluation, and report the clean and robust accuracy of N-FGSM under different combinations of noise level $k$ and step size $\alpha$. Note that when the noise level $k = 0$ N-FGSM recovers plain FGSM, thus, as we increase $\alpha$ it is equivalent to using FGSM with an increased $\epsilon_{\text{train}}$. On the other hand, when $\alpha = 0$, this is equivalent to training with only random noise augmentation. Based on the results reported in Table 1, we make the following observations:

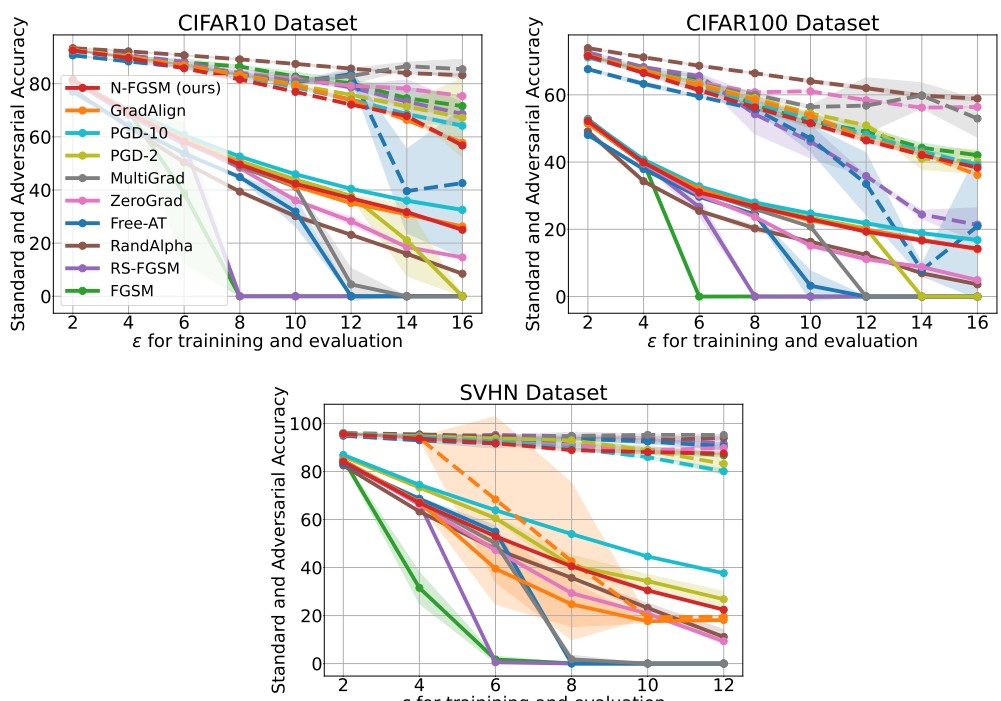

Figure 8: Comparison of all methods on CIFAR-10, CIFAR-100 and SVHN with WideResNet28-10 over different perturbation radius ($\epsilon$ is divided by 255). We plot both the robust (solid line) and the clean (dashed line) accuracy for each method. Legend is shared among all plots.

1) **Increasing the perturbation size is not enough to avoid catastrophic overfitting**. For instance, as observed in the first column of Table 1, training with an increasing $\alpha$ without noise, i.e. ($k = 0$ which is equivalent to FGSM), leads to catastrophic overfitting despite the clear increase in perturbation size due to the increase of $\alpha$.

2) **Catastrophic overfitting leads to a vulnerable model against smaller perturbations too**. For instance, looking at the experiments with $\alpha = 10$ or $\alpha = 12$ in the first column, we observe that the resultant models are vulnerable to adversarial attacks for $\epsilon_{\text{test}} = 8/255$ even though the perturbation radius used in training is larger than $8/255$. This indicates that once a model catastrophically overfits to perturbations of a given radius, it can be vulnerable to smaller perturbations too.

3) **Increasing the level of noise is necessary to stabilize training for larger perturbations**. As we increase the step size $\alpha$ of the attack, we observe that we need to increase the ratio between noise $k$ and step-size $\alpha$ in order to avoid catastrophic overfitting. This again further suggests that it is indeed increasing the perturbation size by increasing the noise level (and not simply increasing the perturbation budget) what mitigates catastrophic overfitting.

4) **Training with noise perturbations has a much milder effect on the clean accuracy than adversarial training**. Despite N-FGSM perturbations have a larger radius, it does not result in a significant drop of the clean accuracy, see Figure 5. Although it might seem counter-intuitive, we note that while N-FGSM has a larger perturbation size, this increase is not only merely in the adversary direction but also in the random noise direction. That is to say, while the perturbation size is larger for N-FGSM, this is not necessarily equivalent to augmenting with adversaries with an a similarly larger perturbation size of adversaries due to the bias in the noise. In Table 1, we observe how augmenting training samples with noise alone (when $\alpha = 0$) has a much milder effect on the clean accuracy. In general, moving right on the table (increasing noise) is more forgiving on the clean accuracy than moving downwards (increasing FGSM step size). This is not so surprising considering that moving in random directions along the input space has a much lower impact on the loss than moving along the FGSM direction (see Figure 12) and that training with noise alone does not provide any significant robustness against larger attacks (for a more detailed ablation see Figure 11).

| | $k = 0$ (no noise) | $k = {}^4\!/255$ ($0.5\epsilon_{\text{train}}$) | $k = {}^8\!/255$ ($1\epsilon_{\text{train}}$) | $k = {}^{16}\!/255$ ($2\epsilon_{\text{train}}$) |
|---|---|---|---|---|
| $\alpha = 0$ | $93.8 \pm 0.14$ | $93.53 \pm 0.12$ | $93.06 \pm 0.02$ | $91.76 \pm 0.07$ |
| | $0.0 \pm 0.0$ | $0.0 \pm 0.0$ | $0.0 \pm 0.0$ | $0.01 \pm 0.0$ |
| $\alpha = 4$ | $88.77 \pm 0.04$ | $88.61 \pm 0.07$ | $88.42 \pm 0.03$ | $87.79 \pm 0.05$ |
| | $35.88 \pm 0.55$ | $36.12 \pm 0.09$ | $36.7 \pm 0.23$ | $37.27 \pm 0.23$ |
| $\alpha = 6$ | $85.58 \pm 0.11$ | $85.52 \pm 0.23$ | $85.03 \pm 0.09$ | $84.49 \pm 0.1$ |
| | $43.85 \pm 0.12$ | $44.14 \pm 0.24$ | $44.44 \pm 0.13$ | $44.44 \pm 0.15$ |
| $\alpha = 8$ | $86.41 \pm 0.7$ | $81.54 \pm 0.19$ | $81.57 \pm 0.07$ | $80.58 \pm 0.22$ |
| | $0.0 \pm 0.0$ | $47.93 \pm 0.11$ | $48.16 \pm 0.21$ | $48.12 \pm 0.07$ |
| $\alpha = 10$ | $82.08 \pm 1.62$ | $82.81 \pm 1.11$ | $77.32 \pm 0.14$ | $76.49 \pm 0.14$ |
| | $0.0 \pm 0.0$ | $0.0 \pm 0.0$ | $49.68 \pm 0.25$ | $49.77 \pm 0.37$ |
| $\alpha = 12$ | $80.6 \pm 2.59$ | $81.75 \pm 1.1$ | $82.0 \pm 1.65$ | $72.52 \pm 0.16$ |
| | $0.0 \pm 0.0$ | $0.0 \pm 0.0$ | $0.0 \pm 0.0$ | $50.17 \pm 0.22$ |

Table 1: Ablation of the clean (top) and PGD-50-10 (bottom) accuracy when changing N-FGSM hyperparameters – noise level $k$ and FGSM step size $\alpha$. Results are averaged over 3 seeds. All models are evaluated with PGD-50-10 attack and $\epsilon_{\text{test}} = \epsilon_{\text{train}} = 8/255$.

## D    ABLATION OF BASELINES INCREASING THE TRAINING EPSILON

We have seen in Theorem 1 and Appendix F that the magnitude of N-FGSM perturbations will (on expectation) be larger than that of other baselines. Moreover, since N-FGSM does not use clipping to ensure perturbations are within the $\epsilon - \ell_\infty$ ball, they could have $\ell_\infty$−norm of up to $k + \alpha$; which for the default hyperparameter values ($\alpha = \epsilon$, $k = 2\epsilon$) add up to $3\epsilon$. In this section, we study the performance of all single-step baselines as we increase the training epsilon.

In Table 2, we present the result of an experiment where we fix $\epsilon_{\text{test}} = {}^8\!/255$, then for each baseline we increase the $\epsilon_{\text{train}}$ from $8/255$ to $16/255$ (while always evaluating the final model at $8/255$). We use the same training hyperparameters as reported in Section 5.1. Results lead to three main observations:

1) **Increasing $\epsilon_{\text{train}}$ ends up hurting test robust accuracy**. Even though in some methods we observe an increase in adversarial accuracy as we start to increase $\epsilon_{\text{train}}$, it ends up decreasing. Moreover, this increase in adversarial accuracy (see GradAlign, MultiGrad or N-FGSM) is at the expense of a decrase in clean accuracy.

2) **Catastrophic overfitting leads to a vulnerable model to smaller perturbations as well**. As observed in Appendix C, those models which suffer catastrophic overfitting become vulnerable to attacks with smaller perturbation radius than used during training. Here, we observe the same effect for various single-step baselines, which indicates this is likely a general trend.

3) **Changing N-FGSM step-size $\alpha$ leads to similar result to changing $\epsilon_{\text{train}}$ in other baselines**. Interestingly, we observe that although the presence of noise will lead to perturbations much larger than those of other methods, it is the step-size in the direction of the FGSM attack that will have a larger impact on robust and clean accuracy. Thus, for larger perturbations the role of noise seems to be mainly to somehow mitigate catastrophic overfitting rather than strongly contributing to model robustness.

In light of the previous observations, we conclude that using a larger perturbation for N-FGSM does not lead to an unfair comparison to other baselines, on the contrary, the best values for these baselines (considering the trade-off between clean and robust accuracy) are when using the same radius of perturbations for training and test ($\epsilon_{\text{test}} = \epsilon_{\text{train}}$). On the other hand, this result highlights the particularity of increasing the perturbation size with noise rather than following the adversarial

| | $\epsilon_{\text{train}} = {}^8/_{255}\ (1\epsilon_{\text{test}})$ | $\epsilon_{\text{train}} = {}^{12}/_{255}\ (1.5\epsilon_{\text{test}})$ | $\epsilon_{\text{train}} = {}^{16}/_{255}\ (2\epsilon_{\text{test}})$ |
|---|---|---|---|
| FGSM | $86.41 \pm 0.7$ | $80.6 \pm 2.59$ | $77.14 \pm 2.46$ |
| | $0.0 \pm 0.0$ | $0.0 \pm 0.0$ | $0.0 \pm 0.0$ |
| RS-FGSM | $84.05 \pm 0.13$ | $65.22 \pm 23.23$ | $76.66 \pm 0.38$ |
| | $46.08 \pm 0.18$ | $0.0 \pm 0.0$ | $0.0 \pm 0.0$ |
| Kim et. al. | $89.02 \pm 0.1$ | $88.35 \pm 0.31$ | $90.45 \pm 0.08$ |
| | $33.01 \pm 0.09$ | $27.36 \pm 0.31$ | $9.28 \pm 0.12$ |
| AT Free | $78.41 \pm 0.18$ | $73.91 \pm 4.19$ | $71.64 \pm 3.89$ |
| | $46.03 \pm 0.36$ | $32.4 \pm 22.91$ | $0.0 \pm 0.0$ |
| ZeroGrad | $82.62 \pm 0.05$ | $78.11 \pm 0.2$ | $75.42 \pm 0.13$ |
| | $47.08 \pm 0.1$ | $46.43 \pm 0.37$ | $45.63 \pm 0.39$ |
| MultiGrad | $82.33 \pm 0.14$ | $75.28 \pm 0.2$ | $71.42 \pm 5.63$ |
| | $47.29 \pm 0.07$ | $50.0 \pm 0.79$ | $16.01 \pm 22.64$ |
| GradAlign | $81.9 \pm 0.22$ | $73.29 \pm 0.23$ | $61.3 \pm 0.15$ |
| | $48.14 \pm 0.15$ | $50.6 \pm 0.45$ | $46.67 \pm 0.29$ |
| N-FGSM | $80.58 \pm 0.22$ | $71.46 \pm 0.14$ | $63.18 \pm 0.49$ |
| | $48.12 \pm 0.07$ | $50.23 \pm 0.31$ | $46.46 \pm 0.1$ |

Table 2: Ablation of the PGD-50-10 accuracy for single-step methods when increasing the $\epsilon_{\text{train}}$. All models are evaluated with PGD-50-10 attack and $\epsilon_{\text{test}} = {}^8/_{255}$. Note that considering the trade-off between clean and robust accuracy, all methods perform best when training with the same epsilon to be applied at test time.

direction. Gaining a deeper understanding of the role of noise in avoiding catastrophic overfitting is a promising direction for future work.

## E  LONGER TRAINING SCHEDULE

In our experiments, we have followed the "fast" training schedule introduced by Wong et al. (2020). However, Rice et al. (2020) suggest that a longer training schedule coupled with early stopping may lead to a boost in performance. We also use the long training schedule for N-FGSM and observe that it does not lead to catastrophic overfitting. In Table 3 we compare the performance of N-FGSM and GradAlign for the long training schedule. We observe that GradAlign does not seem to benefit from the long training schedule. On the other hand, although N-FGSM seems to obtain a slight increase in performance, the "fast" schedule provides comparable performance. It is worth mentioning that for GradAlign, the default regularizer hyperparameter for $\epsilon = {}^8/_{255}$ and CIFAR-10 ($\lambda = 0.2$) does not prevent catastrophic overfitting. We do a hyperparameter search and keep the value with the largest final robust accuracy ($\lambda = 0.632$).

| N-FGSM | | Grad Align | |
|---|---|---|---|
| **Clean Acc** | **Robust Acc** | **Clean Acc** | **Robust Acc** |
| **Long schedule: Final model** | | | |
| $\mathbf{83.18 \pm 0.11}$ | $36.56 \pm 0.26$ | $\mathbf{84.13 \pm 0.24}$ | $36.17 \pm 0.19$ |
| **Long schedule: Best model** | | | |
| $80.8 \pm 0.36$ | $\mathbf{48.48 \pm 0.27}$ | $81.57 \pm 0.44$ | $47.86 \pm 0.1$ |
| **fast schedule: Final model** | | | |
| $80.58 \pm 0.22$ | $48.12 \pm 0.07$ | $81.9 \pm 0.22$ | $\mathbf{48.14 \pm 0.15}$ |

Table 3: Comparison of "long" (Rice et al., 2020) and "fast" (Wong et al., 2020) training schedules for N-FGSM and GradAlign. GradAlign does not seem to benefit from the long training schedule. Although N-FGSM seems to obtain a slight increase in performance, the "fast" schedule provides comparable performance.

## F    MAGNITUDE OF N-FGSM PERTURBATIONS

**Lemma 1** (Expected perturbation). *Consider the N-FGSM perturbation as defined in Equation* (4)
$$\delta_{\text{N-FGSM}} = \eta + \alpha \cdot sign\left(\nabla_x \ell(f(x+\eta), y)\right), \; where \; \eta \sim \Omega.$$

*Let the distribution $\Omega$ be the uniform distribution $\mathcal{U}\left(\left[-k\epsilon, k\epsilon\right]^d\right)$ and $\alpha > 0$. Then,*

$$\mathbb{E}_\eta\left[\|\delta_{\text{N-FGSM}}\|_2^2\right] = d\left(\frac{k^2\epsilon^2}{3} + \alpha^2\right)$$

*and*

$$\mathbb{E}_\eta\left[\|\delta_{\text{N-FGSM}}\|_2\right] \leq \sqrt{d\left(\frac{k^2\epsilon^2}{3} + \alpha^2\right)}$$

*Proof.* By Jensen's inequality, we have

$$\mathbb{E}_\eta\left[\|\delta_{\text{N-FGSM}}\|_2\right] \leq \sqrt{\mathbb{E}_\eta\left[\|\delta_{\text{N-FGSM}}\|_2^2\right]}$$

Then let us consider the term $\mathbb{E}_\eta\left[\|\delta_{\text{N-FGSM}}\|_2^2\right]$ and use the shorthand $\nabla(\eta)_i = \left(\nabla_x\ell(f(x+\eta), y)\right)_i$.

$$
\begin{aligned}
\mathbb{E}_\eta\left[\|\delta_{\text{N-FGSM}}\|_2^2\right] &= \mathbb{E}_\eta \|\eta + \alpha \cdot \text{sign}\left(\nabla_x\ell(f(x+\eta), y)\right)\|_2^2 \\
&= \mathbb{E}_\eta\left[\sum_{i=1}^d \left(\eta_i + \alpha \cdot \text{sign}(\nabla(\eta)_i)\right)^2\right] \\
&= \sum_{i=1}^d \mathbb{E}_\eta\left[\left(\eta_i + \alpha \cdot \text{sign}(\nabla(\eta)_i)\right)^2\right] \\
&= \sum_{i=1}^d \mathbb{E}_\eta\left[\left(\eta_i + \alpha \cdot \text{sign}(\nabla(\eta)_i)\right)^2 |\text{sign}(\nabla(\eta)_i) = 1\right]\mathbb{P}_\eta\left[\text{sign}(\nabla(\eta)_i) = 1\right] \\
&\quad + \sum_{i=1}^d \mathbb{E}_\eta\left[\left(\eta_i + \alpha \cdot \text{sign}(\nabla(\eta)_i)\right)^2 |\text{sign}(\nabla(\eta)_i) = -1\right]\mathbb{P}_\eta\left[\text{sign}(\nabla(\eta)_i) = -1\right] \\
&= \sum_{i=1}^d \frac{1}{2k\epsilon} \int_{-k\epsilon}^{k\epsilon} \left(\eta_i + \alpha\right)^2 d\eta_i \cdot \mathbb{P}_\eta\left[\text{sign}(\nabla(\eta)_i) = 1\right] \\
&\quad + \frac{1}{2k\epsilon}\sum_{i=1}^d \int_{-k\epsilon}^{k\epsilon} \left(\eta_i - \alpha\right)^2 d\eta_i \cdot \mathbb{P}_\eta\left[\text{sign}(\nabla(\eta)_i) = -1\right] \\
&= \sum_{i=1}^d \frac{1}{2k\epsilon} \int_{\alpha-k\epsilon}^{\alpha+k\epsilon} z^2 dz \cdot \mathbb{P}_\eta\left[\text{sign}(\nabla(\eta)_i) = 1\right] \\
&\quad + \frac{1}{2k\epsilon}\sum_{i=1}^d \int_{-\alpha-k\epsilon}^{-\alpha+k\epsilon} z^2 dz \cdot \mathbb{P}_\eta\left[\text{sign}(\nabla(\eta)_i) = -1\right] \\
&= \sum_{i=1}^d \frac{1}{2k\epsilon} \int_{\alpha-k\epsilon}^{\alpha+k\epsilon} z^2 dz \cdot \mathbb{P}_\eta\left[\text{sign}(\nabla(\eta)_i) = 1\right] \\
&\quad + \frac{1}{2k\epsilon}\sum_{i=1}^d \int_{\alpha-k\epsilon}^{\alpha+k\epsilon} z^2 dz \cdot \mathbb{P}_\eta\left[\text{sign}(\nabla(\eta)_i) = -1\right] \\
&= \frac{1}{2k\epsilon} \int_{\alpha-k\epsilon}^{\alpha+k\epsilon} z^2 dz \sum_{i=1}^d \left(\mathbb{P}_\eta\left[\text{sign}(\nabla(\eta)_i) = 1\right] + \mathbb{P}_\eta\left[\text{sign}(\nabla(\eta)_i) = -1\right]\right) \\
&= \frac{d}{6k\epsilon}\left[(\alpha+k\epsilon)^3 - (\alpha-k\epsilon)^3\right] = \frac{dk^2\epsilon^2}{3} + d\alpha^2
\end{aligned}
$$

Therefore,

$$\mathbb{E}_{\eta}\left[\|\delta_{\text{N-FGSM}}\|_2\right] \leq \sqrt{d\left(\frac{k^2\epsilon^2}{3} + \alpha^2\right)}.$$

$\square$

We state again Theorem 1 and present the proof.

**Theorem 1.** *Let $\delta_{\text{N-FGSM}}$ be our proposed single-step method defined by Equation (4), $\delta_{FGSM}$ be the FGSM method (Goodfellow et al., 2015) and $\delta_{RS\text{-}FGSM}$ be the RS-FGSM method (Wong et al., 2020). Then, with default hyperparameter values and for any $\epsilon > 0$, we have that*

$$\mathbb{E}_{\eta}\left[\|\delta_{N\text{-}FGSM}\|_2^2\right] > \mathbb{E}_{\eta}\left[\|\delta_{FGSM}\|_2^2\right] > \mathbb{E}_{\eta}\left[\|\delta_{RS\text{-}FGSM}\|_2^2\right].$$

*Proof.* From Lemma 1 we have that

$$\mathbb{E}_{\eta}\left[\|\delta_{\text{N-FGSM}}\|_2^2\right] = d\left(\frac{k^2\epsilon^2}{3} + \alpha^2\right).$$

On the other hand, Andriushchenko & Flammarion (2020) showed that

$$\mathbb{E}_{\eta}\left[\|\delta_{\text{RS-FGSM}}\|_2^2\right] = d\left(-\frac{1}{6\epsilon}\alpha^3 + \frac{1}{2}\alpha^2 + \frac{1}{3}\epsilon^2\right).$$

Finally, we note that

$$\mathbb{E}_{\eta}\left[\|\delta_{\text{FGSM}}\|_2^2\right] = \|\delta_{\text{FGSM}}\|_2^2 = d\epsilon^2.$$

The default hyperparameters for N-FGSM are $k = 2, \ \alpha = \epsilon$ and RS-FGSM uses $\alpha = 5\epsilon/4$. With these hyperparameters and any $\epsilon > 0$ we have

$$\mathbb{E}_{\eta}\left[\|\delta_{\text{N-FGSM}}\|_2^2\right] = \frac{7}{3}d\epsilon^2 > \mathbb{E}_{\eta}\left[\|\delta_{\text{FGSM}}\|_2^2\right] = d\epsilon^2 > \mathbb{E}_{\eta}\left[\|\delta_{\text{RS-FGSM}}\|_2^2\right] = \frac{101}{128}d\epsilon^2$$

$\square$

In Lemma 1 we compute the expected value of the squared $\ell_2$ norm of N-FGSM perturbations and by Jensen's inequality we obtain an upper bound for the expected $\ell_2$ norm of N-FGSM perturbations. However, obtaining the exact expected magnitude is more complex. To compliment our analytic results, we approximate the $\ell_2$ norm of FGSM, RS-FGSM and N-FGSM via Monte Carlo sampling. Results are presented in Figure 9. We observe that the empirical estimations are very close to the analytical upper bounds and that indeed, N-FGSM has a magnitude significantly above that of FGSM or RS-FGSM.

## G  N-FGSM WITH GAUSSIAN NOISE

In the main paper we have only explored noise sources coming from a Uniform distribution. Since we are measuring robustness against $l_\infty-$ attacks, the Uniform distribution is a natural choice because the random perturbations will be bounded to the $l_\infty$ ball defined by the span of the distribution. However, for the sake of completeness, we also explore the performance of augmenting the samples from a Gaussian distribution where we choose its standard deviation to match that of the uniform distribution. In Table 4 we present a comparison of the clean (top) and PGD-50-10 (bottom) accuracy for different values of $\alpha$ and noise magnitude with $\epsilon = {}^8/_{255}$. Recall that by default we use Uniform distribution $\mathcal{U}[-k, k]$, therefore hyperparameter $k$ sets the noise magnitude.

Increasing the FGSM step size without increasing the amount of noise leads to catastrophic overfitting. Note results for $k = 0.5\epsilon$. More importantly, results are very similar when the two noise distributions share the same standard deviation. Thus, using Gaussian instead of Uniform noise does not seem to alter the results. Although this might be expected, we remark that the Gaussian is an unbounded noise distribution and the common practice in adversarial training is to always restrict the norm of the perturbations.

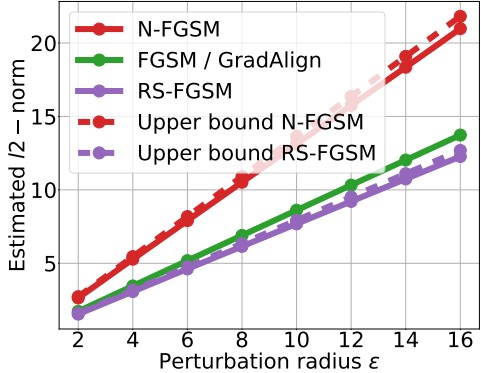

Figure 9: Monte Carlo estimations of the expected $l_2-$norm of perturbations from different methods and corresponding analytical upper bounds. As mentioned in Andriushchenko & Flammarion (2020), we observe that RS-FGSM perturbations have lower $l_2$ norm than FGSM. However, N-FGSM perturbations have a significantly higher $l_2-$norm than both RS-FGSM and FGSM. This seems to indicate that the role of random step is not simply to lower the $l_2$ norm as previously suggested (Andriushchenko & Flammarion, 2020).

| | **Uniform Noise** | | | **Gaussian Noise** | | |
|---|---|---|---|---|---|---|
| | $\alpha = {}^6/_{255}\,(0.75\epsilon)$ | $\alpha = {}^8/_{255}\,(1\epsilon)$ | $\alpha = {}^{10}/_{255}\,(1.25\epsilon)$ | $\alpha = {}^6/_{255}\,(0.75\epsilon)$ | $\alpha = {}^8/_{255}\,(1\epsilon)$ | $\alpha = {}^{10}/_{255}\,(1.25\epsilon)$ |
| $k = 0.5\epsilon$ | $85.52 \pm 0.23$ | $81.54 \pm 0.19$ | $82.81 \pm 1.11$ | $85.27 \pm 0.11$ | $81.71 \pm 0.27$ | $83.34 \pm 1.48$ |
| | $44.14 \pm 0.24$ | $47.93 \pm 0.11$ | $0.0 \pm 0.0$ | $44.23 \pm 0.17$ | $47.98 \pm 0.14$ | $0.0 \pm 0.0$ |
| $k = 1\epsilon$ | $85.03 \pm 0.09$ | $81.57 \pm 0.07$ | $77.32 \pm 0.14$ | $85.01 \pm 0.17$ | $81.35 \pm 0.14$ | $77.22 \pm 0.32$ |
| | $44.44 \pm 0.13$ | $48.16 \pm 0.21$ | $49.68 \pm 0.25$ | $44.41 \pm 0.04$ | $48.21 \pm 0.11$ | $49.83 \pm 0.1$ |
| $k = 2\epsilon$ | $84.49 \pm 0.1$ | $80.58 \pm 0.22$ | $76.49 \pm 0.14$ | $84.35 \pm 0.24$ | $80.44 \pm 0.31$ | $76.33 \pm 0.37$ |
| | $44.44 \pm 0.15$ | $48.12 \pm 0.07$ | $49.77 \pm 0.37$ | $44.59 \pm 0.22$ | $48.34 \pm 0.1$ | $49.77 \pm 0.23$ |

Table 4: Comparison of the clean (top) and PGD-50-10 (bottom) accuracy across different values of step-size $\alpha$ and noise magnitude for the Uniform and Gaussian distributions with $\epsilon = 8/255$. For every value of $k$, we use a Gaussian with matching standard deviation. We observe that when we match the standard deviation, both distribution perform similarly.

## H   FURTHER RESULTS WITH RANDALPHA

In Section 5.2 we analyze the method presented by Kim et al. (2021) and suggest a baseline where, instead of evaluating intermediate points to determine the "optimal" step size, we simply choose it randomly. That is, we multiply the RS-FGSM perturbation by a random scalar sampled from a uniform distribution in $[0, 1]$. Interestingly we find that it outperforms Kim et al. (2021) without the additional cost of intermediate evaluations and without the need to perform hyperparameter selection to find the optimal number of intermediate evaluations.

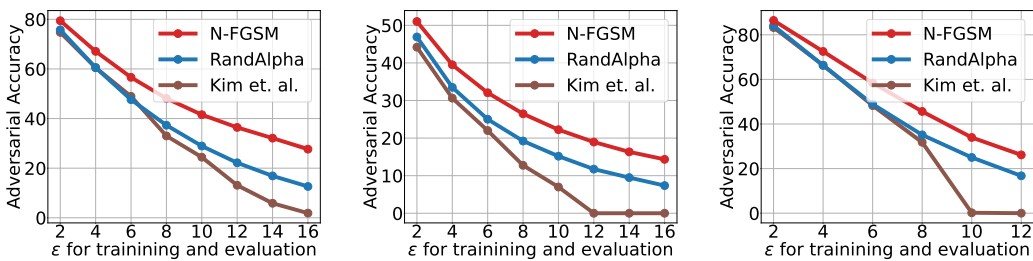

Figure 10: Comparison of Kim et al. (2021) with RandomAlpha, our baseline where we multiply the RS-FGSM perturbation by a scalar uniformly sampled in $[0, 1]$. We present results on CIFAR-10, CIFAR-100 and SVHN with PreActResNet18.

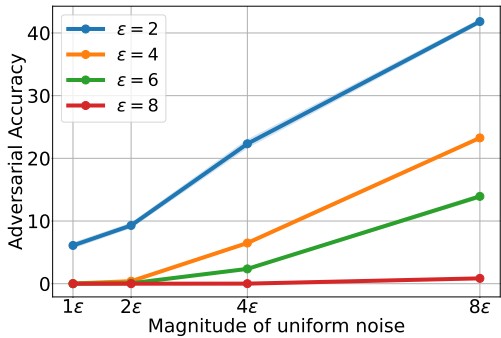

Figure 11: Training with uniform noise augmented samples improves adversarial accuracy for small perturbations but is not effective to protect against larger $l_\infty$ radius $\epsilon$. This motivates us to further augment the noisy samples with FGSM. All experiments are averaged over 3 runs.

## I  TRAINING WITH NOISE AUGMENTED SAMPLES

Gilmer et al. (2019) and Fawzi et al. (2018) report a close link between robustness to adversarial attacks and robustness to random noise. Actually, Gilmer et al. (2019) report that training with noise-augmented samples can improve adversarial accuracy and vice-versa. We note that N-FGSM can actually be seen as a combination of noise-augmentation and adversarial attacks. Here we perform an ablation where we train models with samples augmented with uniform noise $\mathcal{U}[-k, k]$ and then test the PGD-50-10 accuracy. We observe, that indeed random noise can increase the robustness to wort-case perturbations for small $\epsilon - l_\infty$ balls. However, as we increase $\epsilon$, noise augmentation is no longer very effective. However, with N-FGSM, we apply a weak attack to these noise-augmented samples and this seems to be enough to make them effective for adversarial training.

## J  VISUALIZATION OF THE LOSS SURFACE

In this section we present a visualization of the loss surface. We adapted the code from Kim et al. (2021) to analyse the shape of the loss surface at the end of training for different methods. Kim et al. (2021) reported that after adversarial training catastrophic overfitting, the loss surface would become non-linear. In particular, they found that the FGSM perturbation seems to be misguided by local maxima very close to the clean image that result in ineffective attacks. We note this was already reported by Tramèr et al. (2018) which proposed to perform a random step to *escape* those maxima. We argue that adding noise to the random step, when properly implemented, actually prevents those maxima to appear in the first place.

## K  COMPARISON OF ADVERSARIAL TRAINING COST

In this section we describe how we compute the relative training cost for single-step methods shown in Figure 1 (right). We approximate the cost based on the number of forward/backward passes each method uses, disregarding the cost of other additional operations such as adding a random step for RS-FGSM or N-FGSM. We understand these operations have a negligible cost compared to a full forward or backward pass.

**FGSM:** FGSM is the cheapest of all methods since it only uses one forward/backward to compute the attack and an additional forward/backward to compute the weight update. Hence, Cost FGSM = 2 F/B.

**RS-FGSM:** As previously mentioned, we do not take into account the cost of random steps or clipping, hence we consider RS-FGSM to have the same cost as standard FGSM. Cost RS-FGSM = 2 F/B.

**N-FGSM:** Idem as before, cost of N-FGSM = 2 F/B.

**ZeroGrad:** For ZeroGrad they need to do an additional sorting operation to find the smallest gradient components. This could be potentially expensive, however, since the size of the input image is

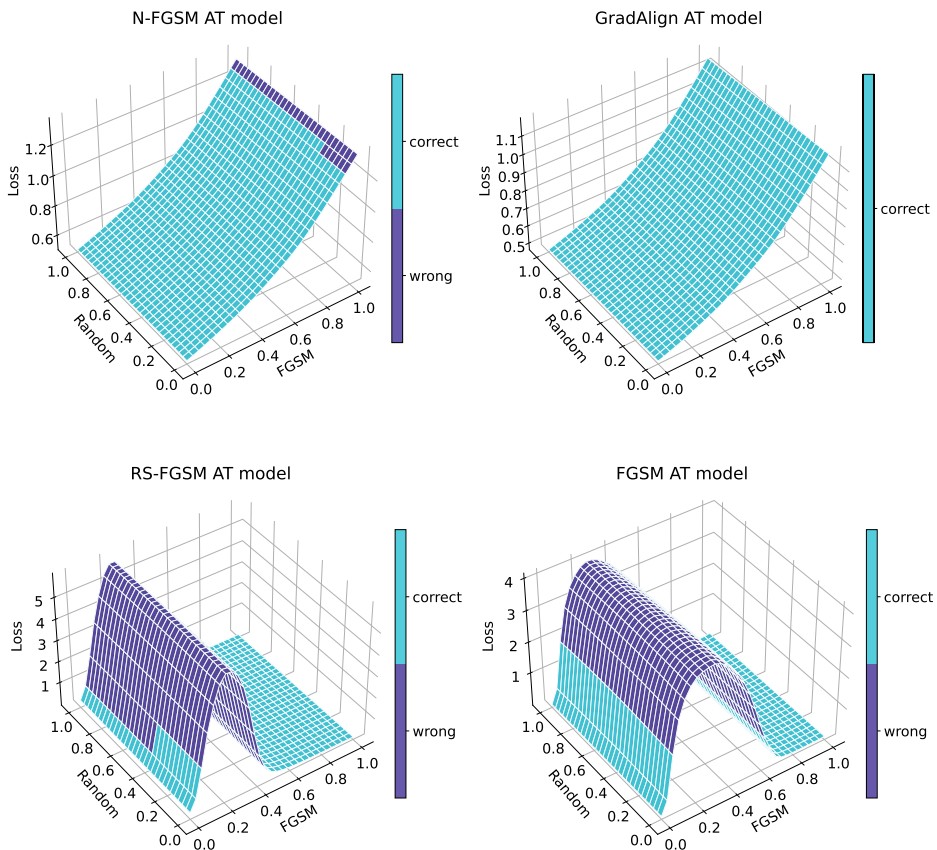

Figure 12: Visualization of the loss surface for models trained using different methods. Given a clean sample from the test set in coordinate $(0, 0)$, we compute the FGSM perturbation and evaluate the loss on the subspace generated by the FGSM perturbation direction and a random direction. That is, we evaluate $x_{\text{clean}} + t_1 \cdot \delta_{\text{FGSM}} + t_2 \cdot \delta_{\text{Random}}$, where $t_1, \ t_2 \in [0, 1]$. Note that FGSM and RS-FGSM both have catastrophic overfitting and the final models present a highly non-linear loss surface, on the other hand, both N-FGSM and GradAlign produce final models with a very linear loss surface which is key to obtain meaningful perturbations.

several orders of magnitude smaller than that of the network, we also ignore this cost. Cost ZeroGrad = 2 F/B.

**MultiGrad:** MultiGrad computes 3 random steps and evaluates the gradient in all of them. Therefore, it needs to do 3 F/B to compute the attack and an additional one to update the weights. Cost MultiGrad = 4 F/B.

**Kim et al. (2021):** Kim et al. (2021) compute the RS-FGSM perturbation and evaluate the model on $c$ points along this direction. Therefore, they will spend 1F/B on the RS-FGSM attack, $c - 1$ F on the evaluations since the clean image has already been evaluated; and 1 F/B for the weight update. In our plot, we used $c = 3$ since it was the most chosen setting. Kim et al. (2021) assume the cost of a forward is similar to that of a backward pass, following this assumption, cost of Kim et al. (2021) is 1 F/B + 2 F + 1 F/B = 3 F/B

**Free-AT:** Shafahi et al. (2019) re-use the gradient from the previous backward pass to compute the FGSM perturbation of the current iteration. Hence, the cost of their training is only 1 F/B per iteration. However, Wong et al. (2020) observed they needed a longer training schedule to produce comparable results. Therefore, the total training cost per iteration (1 F/B) is scaled by 96 in the case of Free-AT, while it is only scaled by 30 for other methods. Relative cost Free = (96 · 1 F/B) / (30 · 2 F/B).

**GradAlign:** Finally, GradAlign uses FGSM with a regularizer. However, this regularizer needs to compute second-order derivatives via double backpropagation, which does not have the same cost

as regular backpropagation. Andriushchenko & Flammarion (2020) report that the cost of using GradAlign regularizer increased the cost of FGSM by 3.

## L    DETAILED RESULTS FOR SECTION 5.1 AND SECTION 5.6

In this section we present the tables with the exact numbers used in plots comparing adversarial training methods. For each method and $\epsilon - l_\infty$ radius, the top number is the clean accuracy while the bottom number is the PGD-50-10 accuracy. We separate single-step from multi-step methods with a double line.

**PreActResNet18 – CIFAR-10 Dataset**

| | $\epsilon = 2/255$ | $\epsilon = 4/255$ | $\epsilon = 6/255$ | $\epsilon = 8/255$ | $\epsilon = 10/255$ | $\epsilon = 12/255$ | $\epsilon = 14/255$ | $\epsilon = 16/255$ |
|---|---|---|---|---|---|---|---|---|
| **N-FGSM** | $91.48 \pm 0.17$ | $88.44 \pm 0.09$ | $84.72 \pm 0.04$ | $80.58 \pm 0.22$ | $75.98 \pm 0.1$ | $71.46 \pm 0.14$ | $67.11 \pm 0.37$ | $63.18 \pm 0.49$ |
| | $\mathbf{79.43 \pm 0.21}$ | $\mathbf{67.09 \pm 0.31}$ | $\mathbf{56.62 \pm 0.26}$ | $\mathbf{48.12 \pm 0.07}$ | $\mathbf{41.56 \pm 0.16}$ | $\mathbf{36.43 \pm 0.16}$ | $\mathbf{32.11 \pm 0.2}$ | $\mathbf{27.67 \pm 0.93}$ |
| Grad Align | $91.73 \pm 0.04$ | $88.76 \pm 0.0$ | $85.67 \pm 0.02$ | $81.9 \pm 0.22$ | $77.54 \pm 0.06$ | $73.29 \pm 0.23$ | $68.01 \pm 0.32$ | $61.3 \pm 0.15$ |
| | $79.16 \pm 0.03$ | $\mathbf{67.13 \pm 0.26}$ | $\mathbf{56.27 \pm 0.31}$ | $\mathbf{48.14 \pm 0.15}$ | $40.75 \pm 0.28$ | $34.51 \pm 0.63$ | $30.36 \pm 0.27$ | $\mathbf{26.64 \pm 0.27}$ |
| FGSM | $91.6 \pm 0.1$ | $88.77 \pm 0.04$ | $85.58 \pm 0.11$ | $86.41 \pm 0.7$ | $82.08 \pm 1.62$ | $80.6 \pm 2.59$ | $76.04 \pm 2.37$ | $77.14 \pm 2.46$ |
| | $79.35 \pm 0.06$ | $67.11 \pm 0.09$ | $56.33 \pm 0.41$ | $0.0 \pm 0.0$ | $0.0 \pm 0.0$ | $0.0 \pm 0.0$ | $0.0 \pm 0.0$ | $0.0 \pm 0.0$ |
| RS-FGSM | $92.09 \pm 0.05$ | $89.69 \pm 0.01$ | $87.0 \pm 0.12$ | $84.05 \pm 0.13$ | $85.21 \pm 0.51$ | $65.22 \pm 23.23$ | $43.59 \pm 25.01$ | $76.66 \pm 0.38$ |
| | $78.64 \pm 0.08$ | $66.12 \pm 0.22$ | $54.87 \pm 0.22$ | $46.08 \pm 0.18$ | $0.0 \pm 0.0$ | $0.0 \pm 0.0$ | $0.0 \pm 0.0$ | $0.0 \pm 0.0$ |
| Kim et. al. | $92.85 \pm 0.11$ | $91.1 \pm 0.04$ | $89.34 \pm 0.05$ | $89.02 \pm 0.1$ | $88.27 \pm 0.14$ | $88.35 \pm 0.31$ | $90.01 \pm 0.25$ | $90.45 \pm 0.08$ |
| | $74.74 \pm 0.35$ | $60.51 \pm 0.4$ | $48.95 \pm 0.45$ | $33.01 \pm 0.09$ | $24.43 \pm 0.84$ | $13.11 \pm 0.63$ | $5.86 \pm 0.57$ | $1.88 \pm 0.05$ |
| AT Free | $87.99 \pm 0.16$ | $84.98 \pm 0.13$ | $81.77 \pm 0.11$ | $78.41 \pm 0.18$ | $74.79 \pm 0.22$ | $73.91 \pm 4.19$ | $61.92 \pm 14.94$ | $71.64 \pm 3.89$ |
| | $74.27 \pm 0.33$ | $62.47 \pm 0.25$ | $53.18 \pm 0.15$ | $46.03 \pm 0.36$ | $39.87 \pm 0.07$ | $22.99 \pm 16.26$ | $0.0 \pm 0.0$ | $0.0 \pm 0.0$ |
| ZeroGrad | $91.71 \pm 0.08$ | $88.8 \pm 0.11$ | $85.71 \pm 0.1$ | $82.62 \pm 0.05$ | $79.91 \pm 0.12$ | $78.11 \pm 0.2$ | $75.66 \pm 0.46$ | $75.42 \pm 0.13$ |
| | $79.36 \pm 0.05$ | $\mathbf{67.32 \pm 0.02}$ | $56.14 \pm 0.21$ | $47.08 \pm 0.1$ | $37.58 \pm 0.2$ | $27.41 \pm 0.27$ | $21.29 \pm 0.97$ | $13.06 \pm 0.22$ |
| MultiGrad | $91.57 \pm 0.16$ | $88.74 \pm 0.12$ | $85.75 \pm 0.05$ | $82.33 \pm 0.14$ | $78.73 \pm 0.16$ | $75.28 \pm 0.2$ | $80.94 \pm 5.94$ | $71.42 \pm 5.63$ |
| | $79.34 \pm 0.02$ | $66.81 \pm 0.02$ | $56.02 \pm 0.3$ | $47.29 \pm 0.07$ | $40.11 \pm 0.24$ | $33.87 \pm 0.17$ | $9.55 \pm 13.5$ | $16.35 \pm 11.57$ |
| PGD-2 | $91.4 \pm 0.07$ | $88.46 \pm 0.13$ | $85.14 \pm 0.13$ | $81.41 \pm 0.05$ | $77.18 \pm 0.15$ | $72.9 \pm 0.26$ | $70.39 \pm 2.71$ | $64.81 \pm 11.58$ |
| | $\mathbf{79.55 \pm 0.15}$ | $67.62 \pm 0.03$ | $57.39 \pm 0.13$ | $49.58 \pm 0.08$ | $43.3 \pm 0.11$ | $38.13 \pm 0.15$ | $22.89 \pm 15.26$ | $9.6 \pm 13.37$ |
| PGD-10 | $91.25 \pm 0.04$ | $88.34 \pm 0.11$ | $84.79 \pm 0.11$ | $80.71 \pm 0.14$ | $76.13 \pm 0.35$ | $71.24 \pm 0.3$ | $66.7 \pm 0.39$ | $62.11 \pm 0.62$ |
| | $\mathbf{79.47 \pm 0.13}$ | $\mathbf{68.29 \pm 0.24}$ | $\mathbf{58.85 \pm 0.18}$ | $\mathbf{51.33 \pm 0.31}$ | $\mathbf{45.02 \pm 0.49}$ | $\mathbf{39.93 \pm 0.5}$ | $\mathbf{36.02 \pm 0.67}$ | $\mathbf{32.22 \pm 0.64}$ |

**PreActResNet18 – CIFAR-100 Dataset**

| | $\epsilon = 2/255$ | $\epsilon = 4/255$ | $\epsilon = 6/255$ | $\epsilon = 8/255$ | $\epsilon = 10/255$ | $\epsilon = 12/255$ | $\epsilon = 14/255$ | $\epsilon = 16/255$ |
|---|---|---|---|---|---|---|---|---|
| **N-FGSM** | $69.12 \pm 0.27$ | $64.0 \pm 0.06$ | $59.53 \pm 0.02$ | $54.9 \pm 0.2$ | $50.6 \pm 0.16$ | $46.06 \pm 0.14$ | $41.67 \pm 0.25$ | $37.91 \pm 0.11$ |
| | $\mathbf{51.02 \pm 0.34}$ | $\mathbf{39.5 \pm 0.12}$ | $\mathbf{32.06 \pm 0.37}$ | $\mathbf{26.46 \pm 0.22}$ | $\mathbf{22.23 \pm 0.17}$ | $\mathbf{18.95 \pm 0.15}$ | $\mathbf{16.33 \pm 0.15}$ | $\mathbf{14.34 \pm 0.07}$ |
| Grad Align | $68.96 \pm 0.15$ | $64.71 \pm 0.16$ | $60.42 \pm 0.23$ | $56.53 \pm 0.31$ | $54.06 \pm 0.44$ | $48.87 \pm 0.32$ | $43.84 \pm 0.14$ | $38.93 \pm 0.21$ |
| | $\mathbf{51.31 \pm 0.12}$ | $\mathbf{39.37 \pm 0.25}$ | $\mathbf{31.91 \pm 0.28}$ | $25.8 \pm 0.14$ | $18.7 \pm 1.92$ | $17.86 \pm 0.04$ | $15.51 \pm 0.16$ | $13.62 \pm 0.19$ |
| FGSM | $69.01 \pm 0.13$ | $64.47 \pm 0.15$ | $63.85 \pm 2.18$ | $53.42 \pm 0.65$ | $45.06 \pm 2.29$ | $46.14 \pm 2.58$ | $41.66 \pm 0.88$ | $44.68 \pm 1.74$ |
| | $51.3 \pm 0.19$ | $39.7 \pm 0.16$ | $10.93 \pm 14.64$ | $0.0 \pm 0.0$ | $0.0 \pm 0.0$ | $0.0 \pm 0.0$ | $0.0 \pm 0.0$ | $0.0 \pm 0.0$ |
| RS-FGSM | $69.83 \pm 0.29$ | $65.9 \pm 0.36$ | $62.15 \pm 0.23$ | $55.26 \pm 6.86$ | $32.33 \pm 12.12$ | $36.07 \pm 2.59$ | $21.52 \pm 5.56$ | $20.38 \pm 6.15$ |
| | $50.13 \pm 0.32$ | $38.36 \pm 0.19$ | $30.82 \pm 0.08$ | $0.01 \pm 0.01$ | $0.0 \pm 0.0$ | $0.0 \pm 0.0$ | $0.0 \pm 0.0$ | $0.0 \pm 0.0$ |
| Kim et. al. | $72.92 \pm 0.41$ | $70.16 \pm 0.07$ | $67.98 \pm 0.19$ | $68.07 \pm 0.1$ | $68.37 \pm 0.21$ | $74.09 \pm 0.06$ | $74.06 \pm 0.34$ | $74.01 \pm 0.36$ |
| | $44.19 \pm 0.25$ | $30.63 \pm 0.28$ | $22.0 \pm 0.02$ | $12.75 \pm 0.21$ | $6.98 \pm 0.23$ | $0.0 \pm 0.0$ | $0.0 \pm 0.0$ | $0.0 \pm 0.0$ |
| AT Free | $63.01 \pm 0.19$ | $59.41 \pm 0.27$ | $55.43 \pm 0.37$ | $51.91 \pm 0.08$ | $48.11 \pm 0.09$ | $43.48 \pm 1.25$ | $18.33 \pm 4.86$ | $20.43 \pm 11.25$ |
| | $45.7 \pm 0.33$ | $35.95 \pm 0.09$ | $29.37 \pm 0.21$ | $24.32 \pm 0.4$ | $20.64 \pm 0.22$ | $5.71 \pm 8.05$ | $0.0 \pm 0.0$ | $0.0 \pm 0.0$ |
| ZeroGrad | $69.35 \pm 0.36$ | $64.59 \pm 0.32$ | $60.69 \pm 0.09$ | $56.94 \pm 0.13$ | $54.55 \pm 0.17$ | $52.97 \pm 0.34$ | $50.87 \pm 0.26$ | $50.73 \pm 0.3$ |
| | $\mathbf{51.1 \pm 0.09}$ | $\mathbf{39.38 \pm 0.15}$ | $31.72 \pm 0.21$ | $25.87 \pm 0.09$ | $19.49 \pm 0.08$ | $14.32 \pm 0.08$ | $10.92 \pm 0.59$ | $7.3 \pm 0.16$ |
| MultiGrad | $69.01 \pm 0.16$ | $64.44 \pm 0.11$ | $60.65 \pm 0.26$ | $56.84 \pm 0.2$ | $53.62 \pm 0.25$ | $53.05 \pm 1.85$ | $48.28 \pm 0.66$ | $45.28 \pm 11.14$ |
| | $51.15 \pm 0.03$ | $39.16 \pm 0.03$ | $31.73 \pm 0.09$ | $25.96 \pm 0.11$ | $21.37 \pm 0.16$ | $9.57 \pm 7.32$ | $3.2 \pm 4.49$ | $0.0 \pm 0.0$ |
| PGD-2 | $69.18 \pm 0.1$ | $64.32 \pm 0.14$ | $60.21 \pm 0.13$ | $55.8 \pm 0.16$ | $51.68 \pm 0.1$ | $48.2 \pm 0.1$ | $46.14 \pm 1.24$ | $37.97 \pm 10.52$ |
| | $\mathbf{51.36 \pm 0.03}$ | $40.06 \pm 0.14$ | $32.99 \pm 0.24$ | $27.38 \pm 0.16$ | $23.39 \pm 0.19$ | $19.83 \pm 0.29$ | $10.55 \pm 7.51$ | $4.79 \pm 6.75$ |
| PGD-10 | $68.83 \pm 0.07$ | $63.87 \pm 0.09$ | $59.37 \pm 0.07$ | $54.79 \pm 0.38$ | $50.53 \pm 0.15$ | $46.05 \pm 0.21$ | $41.76 \pm 0.07$ | $37.81 \pm 0.14$ |
| | $\mathbf{51.51 \pm 0.27}$ | $\mathbf{40.59 \pm 0.36}$ | $\mathbf{33.65 \pm 0.02}$ | $\mathbf{28.55 \pm 0.27}$ | $\mathbf{24.17 \pm 0.12}$ | $\mathbf{21.2 \pm 0.12}$ | $\mathbf{18.72 \pm 0.06}$ | $\mathbf{16.59 \pm 0.16}$ |

**PreActResNet18 – SVHN Dataset**

| | $\epsilon = 2/255$ | $\epsilon = 4/255$ | $\epsilon = 6/255$ | $\epsilon = 8/255$ | $\epsilon = 10/255$ | $\epsilon = 12/255$ |
|---|---|---|---|---|---|---|
| **N-FGSM** | $96.01 \pm 0.04$ | $94.54 \pm 0.15$ | $92.25 \pm 0.33$ | $89.56 \pm 0.49$ | $86.74 \pm 0.86$ | $81.48 \pm 1.64$ |
| | $\mathbf{86.44 \pm 0.1}$ | $\mathbf{72.53 \pm 0.19}$ | $58.42 \pm 0.14$ | $\mathbf{45.63 \pm 0.11}$ | $\mathbf{33.96 \pm 0.49}$ | $\mathbf{26.13 \pm 0.81}$ |
| Grad Align | $96.02 \pm 0.05$ | $94.56 \pm 0.21$ | $92.53 \pm 0.24$ | $90.1 \pm 0.34$ | $87.23 \pm 0.75$ | $84.01 \pm 0.46$ |
| | $\mathbf{86.43 \pm 0.1}$ | $72.12 \pm 0.19$ | $57.34 \pm 0.24$ | $43.85 \pm 0.14$ | $32.87 \pm 0.33$ | $23.62 \pm 0.41$ |
| FGSM | $96.04 \pm 0.07$ | $95.67 \pm 0.07$ | $93.73 \pm 0.68$ | $91.74 \pm 0.86$ | $90.76 \pm 0.63$ | $87.17 \pm 0.43$ |
| | $\mathbf{86.5 \pm 0.05}$ | $13.61 \pm 5.83$ | $0.56 \pm 0.72$ | $0.26 \pm 0.36$ | $0.07 \pm 0.1$ | $0.0 \pm 0.0$ |
| RS-FGSM | $96.18 \pm 0.11$ | $95.09 \pm 0.09$ | $95.11 \pm 0.44$ | $94.46 \pm 0.16$ | $93.88 \pm 0.24$ | $92.74 \pm 0.5$ |
| | $86.16 \pm 0.14$ | $71.28 \pm 0.4$ | $0.11 \pm 0.08$ | $0.0 \pm 0.0$ | $0.0 \pm 0.0$ | $0.0 \pm 0.0$ |
| Kim et. al. | $96.35 \pm 0.02$ | $95.25 \pm 0.08$ | $94.83 \pm 0.02$ | $94.88 \pm 0.29$ | $96.61 \pm 0.09$ | $96.61 \pm 0.01$ |
| | $83.26 \pm 0.24$ | $66.32 \pm 0.63$ | $48.27 \pm 0.52$ | $31.8 \pm 1.1$ | $0.18 \pm 0.21$ | $0.0 \pm 0.0$ |
| AT Free | $95.01 \pm 0.09$ | $93.66 \pm 0.12$ | $91.72 \pm 0.29$ | $91.29 \pm 4.07$ | $91.86 \pm 3.66$ | $92.36 \pm 1.0$ |
| | $84.55 \pm 0.27$ | $71.61 \pm 0.75$ | $\mathbf{59.31 \pm 1.0}$ | $0.01 \pm 0.0$ | $0.0 \pm 0.0$ | $0.0 \pm 0.0$ |
| ZeroGrad | $96.06 \pm 0.03$ | $94.81 \pm 0.16$ | $93.53 \pm 0.26$ | $92.42 \pm 1.29$ | $90.34 \pm 0.32$ | $88.09 \pm 0.4$ |
| | $\mathbf{86.43 \pm 0.1}$ | $71.59 \pm 0.22$ | $51.72 \pm 0.53$ | $35.93 \pm 2.73$ | $21.34 \pm 0.31$ | $14.14 \pm 0.32$ |
| MultiGrad | $96.01 \pm 0.08$ | $94.71 \pm 0.17$ | $95.75 \pm 0.58$ | $94.86 \pm 0.97$ | $94.7 \pm 0.12$ | $94.48 \pm 0.19$ |
| | $\mathbf{86.4 \pm 0.08}$ | $\mathbf{71.98 \pm 0.26}$ | $28.1 \pm 18.85$ | $11.49 \pm 16.19$ | $0.0 \pm 0.0$ | $0.0 \pm 0.0$ |
| PGD-2 | $96.03 \pm 0.14$ | $94.66 \pm 0.1$ | $93.77 \pm 0.61$ | $94.63 \pm 1.29$ | $84.09 \pm 14.99$ | $94.16 \pm 0.54$ |
| | $86.72 \pm 0.06$ | $73.29 \pm 0.29$ | $60.53 \pm 0.73$ | $20.68 \pm 18.56$ | $0.41 \pm 0.29$ | $0.02 \pm 0.03$ |
| PGD-10 | $95.92 \pm 0.08$ | $94.37 \pm 0.13$ | $92.46 \pm 0.25$ | $89.67 \pm 0.34$ | $85.75 \pm 0.65$ | $80.08 \pm 0.93$ |
| | $\mathbf{86.94 \pm 0.14}$ | $\mathbf{74.76 \pm 0.19}$ | $\mathbf{63.9 \pm 0.48}$ | $\mathbf{53.95 \pm 0.55}$ | $\mathbf{44.91 \pm 0.45}$ | $\mathbf{37.65 \pm 0.53}$ |

**WideResNet28-10 – CIFAR-10 Dataset**

| | $\epsilon = 2/255$ | $\epsilon = 4/255$ | $\epsilon = 6/255$ | $\epsilon = 8/255$ | $\epsilon = 10/255$ | $\epsilon = 12/255$ | $\epsilon = 14/255$ | $\epsilon = 16/255$ |
|---|---|---|---|---|---|---|---|---|
| **N-FGSM** | $92.51 \pm 0.11$ | $89.65 \pm 0.09$ | $85.8 \pm 0.23$ | $81.59 \pm 0.32$ | $76.92 \pm 0.04$ | $72.13 \pm 0.15$ | $67.82 \pm 0.43$ | $56.73 \pm 0.42$ |
| | $\mathbf{81.43 \pm 0.3}$ | $69.11 \pm 0.24$ | $58.29 \pm 0.14$ | $49.53 \pm 0.25$ | $\mathbf{42.37 \pm 0.36}$ | $\mathbf{36.85 \pm 0.2}$ | $\mathbf{31.66 \pm 0.6}$ | $25.01 \pm 0.23$ |
| Grad Align | $92.59 \pm 0.05$ | $89.95 \pm 0.3$ | $86.98 \pm 0.06$ | $83.19 \pm 0.26$ | $79.35 \pm 0.26$ | $73.79 \pm 0.72$ | $66.38 \pm 0.53$ | $57.75 \pm 0.75$ |
| | $\mathbf{81.33 \pm 0.4}$ | $\mathbf{69.81 \pm 0.47}$ | $\mathbf{59.0 \pm 0.13}$ | $\mathbf{50.0 \pm 0.05}$ | $41.48 \pm 0.51$ | $35.06 \pm 0.74$ | $30.83 \pm 0.39$ | $\mathbf{26.26 \pm 0.13}$ |
| FGSM | $92.65 \pm 0.17$ | $90.06 \pm 0.18$ | $87.99 \pm 1.3$ | $86.46 \pm 0.45$ | $82.67 \pm 1.78$ | $80.14 \pm 1.2$ | $74.54 \pm 4.01$ | $71.56 \pm 3.78$ |
| | $\mathbf{81.38 \pm 0.22}$ | $\mathbf{69.59 \pm 0.25}$ | $38.69 \pm 26.54$ | $0.0 \pm 0.0$ | $0.0 \pm 0.0$ | $0.0 \pm 0.0$ | $0.0 \pm 0.0$ | $0.0 \pm 0.0$ |
| RS-FGSM | $92.85 \pm 0.1$ | $90.73 \pm 0.2$ | $88.24 \pm 0.19$ | $83.64 \pm 1.74$ | $82.1 \pm 1.45$ | $78.62 \pm 0.7$ | $73.25 \pm 8.16$ | $68.64 \pm 4.3$ |
| | $80.9 \pm 0.13$ | $68.23 \pm 0.17$ | $57.21 \pm 0.17$ | $0.0 \pm 0.0$ | $0.0 \pm 0.0$ | $0.0 \pm 0.0$ | $0.0 \pm 0.0$ | $0.0 \pm 0.0$ |
| RandAlpha | $93.37 \pm 0.22$ | $92.17 \pm 0.21$ | $90.71 \pm 0.14$ | $89.16 \pm 0.19$ | $87.44 \pm 0.31$ | $85.69 \pm 0.28$ | $83.98 \pm 0.24$ | $83.23 \pm 0.46$ |
| | $77.67 \pm 0.66$ | $63.73 \pm 0.31$ | $50.4 \pm 0.14$ | $39.37 \pm 0.42$ | $30.13 \pm 0.9$ | $23.13 \pm 0.33$ | $16.0 \pm 0.22$ | $8.47 \pm 0.66$ |
| AT Free | $90.66 \pm 0.25$ | $88.37 \pm 0.15$ | $86.11 \pm 0.29$ | $83.5 \pm 0.27$ | $80.52 \pm 0.32$ | $83.59 \pm 1.35$ | $39.58 \pm 15.8$ | $42.59 \pm 27.96$ |
| | $77.0 \pm 0.27$ | $64.25 \pm 0.33$ | $53.76 \pm 0.48$ | $44.85 \pm 0.39$ | $31.87 \pm 5.53$ | $0.0 \pm 0.0$ | $0.0 \pm 0.0$ | $0.0 \pm 0.0$ |
| ZeroGrad | $92.62 \pm 0.11$ | $90.17 \pm 0.05$ | $86.98 \pm 0.28$ | $84.25 \pm 0.28$ | $81.72 \pm 0.29$ | $79.24 \pm 0.82$ | $78.14 \pm 0.46$ | $75.34 \pm 0.12$ |
| | $\mathbf{81.42 \pm 0.28}$ | $69.28 \pm 0.29$ | $58.4 \pm 0.14$ | $48.29 \pm 0.16$ | $36.08 \pm 0.29$ | $28.24 \pm 1.79$ | $18.54 \pm 0.31$ | $14.6 \pm 0.12$ |
| MultiGrad | $92.64 \pm 0.1$ | $90.18 \pm 0.13$ | $87.11 \pm 0.36$ | $83.87 \pm 0.46$ | $80.89 \pm 0.14$ | $82.88 \pm 2.85$ | $86.6 \pm 1.52$ | $85.46 \pm 3.73$ |
| | $\mathbf{81.19 \pm 0.28}$ | $69.3 \pm 0.2$ | $57.98 \pm 0.08$ | $48.74 \pm 0.09$ | $41.22 \pm 0.57$ | $4.46 \pm 6.09$ | $0.0 \pm 0.0$ | $0.0 \pm 0.0$ |
| PGD-2 | $92.69 \pm 0.14$ | $90.18 \pm 0.19$ | $86.87 \pm 0.18$ | $83.31 \pm 0.16$ | $79.61 \pm 0.47$ | $75.81 \pm 0.24$ | $71.41 \pm 1.38$ | $67.2 \pm 14.94$ |
| | $\mathbf{81.54 \pm 0.18}$ | $69.87 \pm 0.26$ | $59.4 \pm 0.19$ | $50.88 \pm 0.16$ | $43.94 \pm 0.24$ | $37.77 \pm 0.57$ | $21.06 \pm 13.39$ | $0.0 \pm 0.0$ |
| PGD-10 | $92.24 \pm 0.31$ | $89.65 \pm 0.33$ | $86.91 \pm 0.51$ | $82.82 \pm 0.7$ | $78.63 \pm 0.66$ | $74.0 \pm 0.67$ | $68.6 \pm 0.58$ | $64.17 \pm 0.72$ |
| | $81.18 \pm 0.57$ | $\mathbf{70.34 \pm 0.26}$ | $\mathbf{60.59 \pm 0.21}$ | $\mathbf{52.58 \pm 0.2}$ | $\mathbf{45.92 \pm 0.38}$ | $\mathbf{40.44 \pm 0.17}$ | $\mathbf{35.98 \pm 0.56}$ | $\mathbf{32.5 \pm 0.61}$ |

**WideResNet28-10 – CIFAR-100 Dataset**

| | $\epsilon = 2/255$ | $\epsilon = 5/255$ | $\epsilon = 6/255$ | $\epsilon = 8/255$ | $\epsilon = 10/255$ | $\epsilon = 12/255$ | $\epsilon = 14/255$ | $\epsilon = 16/255$ |
|---|---|---|---|---|---|---|---|---|
| **N-FGSM** | $71.56 \pm 0.13$ | $66.49 \pm 0.46$ | $61.38 \pm 0.68$ | $56.23 \pm 0.59$ | $51.54 \pm 0.63$ | $46.43 \pm 0.61$ | $42.11 \pm 0.32$ | $38.34 \pm 0.47$ |
| | $\mathbf{52.23 \pm 0.33}$ | $\mathbf{39.93 \pm 0.37}$ | $30.97 \pm 0.21$ | $\mathbf{26.77 \pm 0.65}$ | $\mathbf{23.03 \pm 0.54}$ | $\mathbf{19.3 \pm 0.59}$ | $\mathbf{16.67 \pm 0.4}$ | $\mathbf{14.27 \pm 0.33}$ |
| Grad Align | $71.68 \pm 0.33$ | $67.09 \pm 0.19$ | $62.86 \pm 0.1$ | $58.55 \pm 0.41$ | $53.85 \pm 0.73$ | $46.94 \pm 0.86$ | $42.63 \pm 0.5$ | $36.17 \pm 0.45$ |
| | $51.5 \pm 0.45$ | $\mathbf{39.9 \pm 0.42}$ | $\mathbf{32.0 \pm 0.22}$ | $26.9 \pm 0.62$ | $22.63 \pm 0.62$ | $\mathbf{19.9 \pm 0.65}$ | $\mathbf{16.93 \pm 0.12}$ | $14.03 \pm 0.24$ |
| FGSM | $71.92 \pm 0.33$ | $67.34 \pm 0.36$ | $64.72 \pm 1.12$ | $56.87 \pm 1.24$ | $52.31 \pm 2.11$ | $48.99 \pm 1.17$ | $44.27 \pm 1.4$ | $42.05 \pm 1.03$ |
| | $\mathbf{52.83 \pm 0.37}$ | $\mathbf{39.83 \pm 0.31}$ | $0.0 \pm 0.0$ | $0.03 \pm 0.05$ | $0.0 \pm 0.0$ | $0.0 \pm 0.0$ | $0.0 \pm 0.0$ | $0.0 \pm 0.0$ |
| RS-FGSM | $72.65 \pm 0.28$ | $68.26 \pm 0.2$ | $65.58 \pm 0.69$ | $54.25 \pm 5.85$ | $46.08 \pm 4.87$ | $35.84 \pm 0.17$ | $24.4 \pm 1.25$ | $21.37 \pm 5.04$ |
| | $51.63 \pm 0.52$ | $39.57 \pm 0.09$ | $26.63 \pm 2.8$ | $0.0 \pm 0.0$ | $0.0 \pm 0.0$ | $0.0 \pm 0.0$ | $0.0 \pm 0.0$ | $0.0 \pm 0.0$ |
| RandAlpha | $73.9 \pm 0.15$ | $71.17 \pm 0.12$ | $68.65 \pm 0.22$ | $66.42 \pm 0.13$ | $64.05 \pm 0.5$ | $61.99 \pm 0.6$ | $59.74 \pm 0.57$ | $58.9 \pm 0.78$ |
| | $49.13 \pm 0.91$ | $34.3 \pm 0.54$ | $25.5 \pm 0.33$ | $20.27 \pm 0.98$ | $16.3 \pm 0.14$ | $12.4 \pm 0.29$ | $6.93 \pm 0.19$ | $3.63 \pm 0.12$ |
| AT Free | $67.62 \pm 0.24$ | $63.27 \pm 0.72$ | $59.53 \pm 0.31$ | $55.77 \pm 0.28$ | $47.02 \pm 3.83$ | $33.52 \pm 9.24$ | $7.87 \pm 1.78$ | $20.92 \pm 21.48$ |
| | $48.07 \pm 0.31$ | $37.93 \pm 0.69$ | $29.7 \pm 0.51$ | $24.43 \pm 0.37$ | $3.23 \pm 4.43$ | $0.0 \pm 0.0$ | $0.0 \pm 0.0$ | $0.0 \pm 0.0$ |
| ZeroGrad | $71.68 \pm 0.07$ | $67.2 \pm 0.14$ | $63.69 \pm 0.14$ | $60.77 \pm 0.26$ | $61.05 \pm 0.38$ | $58.39 \pm 0.16$ | $56.19 \pm 0.11$ | $56.38 \pm 0.18$ |
| | $\mathbf{52.63 \pm 0.61}$ | $39.57 \pm 0.33$ | $30.27 \pm 0.54$ | $23.7 \pm 0.08$ | $15.1 \pm 0.49$ | $11.13 \pm 0.68$ | $8.8 \pm 0.36$ | $4.9 \pm 0.36$ |
| MultiGrad | $71.8 \pm 0.15$ | $67.73 \pm 0.48$ | $63.24 \pm 0.33$ | $60.05 \pm 0.79$ | $56.39 \pm 0.49$ | $56.79 \pm 8.27$ | $59.8 \pm 3.77$ | $52.96 \pm 5.58$ |
| | $51.9 \pm 0.29$ | $39.7 \pm 0.37$ | $31.5 \pm 0.62$ | $26.03 \pm 0.09$ | $20.8 \pm 0.29$ | $0.0 \pm 0.0$ | $0.0 \pm 0.0$ | $0.0 \pm 0.0$ |
| PGD-2 | $71.62 \pm 0.15$ | $67.25 \pm 0.43$ | $63.18 \pm 0.36$ | $59.02 \pm 0.4$ | $54.47 \pm 0.45$ | $50.91 \pm 0.35$ | $41.03 \pm 3.18$ | $40.13 \pm 3.66$ |
| | $51.73 \pm 0.48$ | $\mathbf{40.27 \pm 0.7}$ | $32.23 \pm 0.19$ | $27.13 \pm 0.37$ | $23.43 \pm 0.31$ | $20.23 \pm 0.39$ | $0.03 \pm 0.05$ | $0.0 \pm 0.0$ |
| PGD-10 | $71.11 \pm 0.62$ | $66.9 \pm 0.57$ | $62.05 \pm 0.47$ | $57.64 \pm 0.81$ | $52.84 \pm 0.88$ | $48.14 \pm 0.73$ | $43.14 \pm 0.87$ | $39.2 \pm 0.62$ |
| | $\mathbf{52.5 \pm 0.59}$ | $\mathbf{40.73 \pm 0.56}$ | $\mathbf{32.8 \pm 0.29}$ | $\mathbf{27.97 \pm 0.59}$ | $\mathbf{24.7 \pm 0.36}$ | $\mathbf{21.8 \pm 0.57}$ | $\mathbf{18.87 \pm 0.6}$ | $\mathbf{16.8 \pm 0.57}$ |

**WideResNet28-10 – SVHN Dataset**

| | $\epsilon = 2/255$ | $\epsilon = 4/255$ | $\epsilon = 6/255$ | $\epsilon = 8/255$ | $\epsilon = 10/255$ | $\epsilon = 12/255$ |
|---|---|---|---|---|---|---|
| **N-FGSM** | $95.64 \pm 0.09$ | $93.66 \pm 0.41$ | $91.77 \pm 0.42$ | $88.89 \pm 0.58$ | $88.07 \pm 0.59$ | $87.52 \pm 0.49$ |
| | $\mathbf{84.1 \pm 0.73}$ | $66.9 \pm 0.86$ | $\mathbf{53.0 \pm 0.36}$ | $\mathbf{40.5 \pm 0.37}$ | $\mathbf{30.47 \pm 0.76}$ | $\mathbf{22.43 \pm 0.53}$ |
| Grad Align | $95.41 \pm 0.06$ | $93.9 \pm 0.48$ | $68.36 \pm 34.49$ | $42.62 \pm 32.73$ | $19.3 \pm 0.21$ | $19.53 \pm 0.08$ |
| | $\mathbf{84.57 \pm 0.56}$ | $67.27 \pm 0.54$ | $39.53 \pm 14.89$ | $24.7 \pm 9.34$ | $17.63 \pm 0.62$ | $18.13 \pm 0.52$ |
| FGSM | $95.83 \pm 0.1$ | $95.0 \pm 0.24$ | $94.23 \pm 0.79$ | $91.11 \pm 1.36$ | $88.83 \pm 1.71$ | $86.74 \pm 0.7$ |
| | $\mathbf{85.03 \pm 0.37}$ | $31.53 \pm 6.57$ | $1.7 \pm 1.36$ | $0.13 \pm 0.19$ | $0.0 \pm 0.0$ | $0.0 \pm 0.0$ |
| RS-FGSM | $95.81 \pm 0.25$ | $94.53 \pm 0.4$ | $95.23 \pm 0.26$ | $94.68 \pm 0.62$ | $93.9 \pm 0.52$ | $91.64 \pm 2.98$ |
| | $83.8 \pm 0.43$ | $66.67 \pm 0.65$ | $0.53 \pm 0.26$ | $0.0 \pm 0.0$ | $0.0 \pm 0.0$ | $0.0 \pm 0.0$ |
| RandAlpha | $96.02 \pm 0.23$ | $95.47 \pm 0.18$ | $94.69 \pm 0.26$ | $93.72 \pm 0.44$ | $93.08 \pm 1.45$ | $93.96 \pm 0.68$ |
| | $82.5 \pm 0.45$ | $63.33 \pm 0.53$ | $47.7 \pm 0.99$ | $35.73 \pm 0.34$ | $23.17 \pm 1.97$ | $11.1 \pm 3.05$ |
| AT Free | $94.85 \pm 0.39$ | $92.95 \pm 0.65$ | $91.62 \pm 1.93$ | $93.74 \pm 0.69$ | $92.47 \pm 0.97$ | $90.5 \pm 1.41$ |
| | $83.13 \pm 0.17$ | $\mathbf{68.67 \pm 0.53}$ | $\mathbf{54.93 \pm 2.58}$ | $0.03 \pm 0.05$ | $0.0 \pm 0.0$ | $0.0 \pm 0.0$ |
| ZeroGrad | $95.78 \pm 0.21$ | $94.06 \pm 0.52$ | $92.13 \pm 0.98$ | $91.04 \pm 0.4$ | $88.85 \pm 0.92$ | $89.8 \pm 1.36$ |
| | $\mathbf{84.47 \pm 0.83}$ | $66.1 \pm 0.37$ | $47.3 \pm 0.62$ | $29.33 \pm 0.56$ | $20.77 \pm 0.63$ | $9.33 \pm 0.76$ |
| MultiGrad | $95.63 \pm 0.16$ | $94.27 \pm 0.38$ | $93.64 \pm 1.21$ | $94.83 \pm 1.55$ | $95.26 \pm 0.34$ | $95.22 \pm 0.15$ |
| | $\mathbf{84.37 \pm 0.59}$ | $67.27 \pm 0.31$ | $50.1 \pm 0.9$ | $1.77 \pm 1.72$ | $0.0 \pm 0.0$ | $0.0 \pm 0.0$ |
| PGD-2 | $95.88 \pm 0.35$ | $94.66 \pm 0.1$ | $93.77 \pm 0.61$ | $92.99 \pm 1.11$ | $88.81 \pm 0.93$ | $83.17 \pm 4.78$ |
| | $\mathbf{86.25 \pm 0.7}$ | $73.29 \pm 0.25$ | $60.53 \pm 0.72$ | $40.77 \pm 4.39$ | $34.33 \pm 2.76$ | $26.8 \pm 3.31$ |
| PGD-10 | $95.92 \pm 0.08$ | $94.36 \pm 0.13$ | $92.46 \pm 0.25$ | $89.67 \pm 0.34$ | $85.98 \pm 0.59$ | $80.08 \pm 0.93$ |
| | $\mathbf{86.94 \pm 0.13}$ | $\mathbf{74.46 \pm 0.54}$ | $\mathbf{63.87 \pm 0.49}$ | $\mathbf{53.95 \pm 0.55}$ | $\mathbf{44.59 \pm 0.14}$ | $\mathbf{37.64 \pm 0.49}$ |

