# OpenReview forum: "Towards fast and effective single-step adversarial training"
_ICLR.cc/2022/Conference — ICLR 2022 Submitted_

### Official Review · Reviewer_Gq3c · 2021-11-01

**Correctness:** 3
**Technical Novelty And Significance:** 1
**Empirical Novelty And Significance:** 2
**Recommendation:** 3
**Confidence:** 4

**Main Review:**

Strength:
1. The authors conducted many experiments to verify the effectiveness of their proposed N-FGSM. They presented their results through tables and figures.
2. The experimental results indicate that  the proposed N-FGSM can achieve better performance than RS-FGSM and GradAlign in terms of adversarial accuracy(Robustness).

Weakness:
1. The contribution of this paper is limited. This paper provides little insight as a research paper as the two improvements of N-FGSM are empirical and heuristic to some extent.
2. The motivation of this paper is unclear. The authors raised the CO problem but turn to the two "key components". The two so-called "key components" of adversarial training, clipping and random steps, are unconvincing. It is more like achieving the improvements first and then finding the motivations to fit them in.
3. The evaluation metric is unfair. By eq(4), the adversarial perturbation will be definitely larger than the baselines. Theorem 1 is obvious and hard to be the contribution of this paper. The magnitude of the $\epsilon$ for the baselines should be triple.
4. To my understanding, increasing the magnitude of the adversarial perturbation may lead to a worse clean accuracy. Figure 5 verifies the clean accuracy drop. However, the table in Figure 4 seems to illustrate that the clean accuracy of N-FGSM is maintained compared with the baselines. And the table is very hard to understand.
5. What is the $\epsilon$ axis in Figure 3 and 4? Does it represent the magnitude of $\epsilon$ for training? If yes, why the robustness decreases as the $\epsilon$ increases? If the $\epsilon$ is the magnitude of the evaluation, then PGD-50-10 usually represents a PGD attack with 50 steps and 10/255 l-infi norm.  It isn't very clear here.

**Summary Of The Paper:**

This paper proposes an adversarial training method, noise-FGSM(N-FGSM), which improves the RS-FGSM by removing the clipping perturbation and increasing the magnitude of adding noise. The training speed of N-FGSM is the same as the RS-FGSM, while the experimental results indicate that N-FGSM outperforms RS-FGSM.

**Summary Of The Review:**

To summary, this paper is more like an experiments report instead of a research paper. The contribution is limited, and the evaluation metric seems to be unfair. I suggest a clear rejection for this paper.

---

> ### Author Response · Authors · 2021-11-14
> **Response 1 of 2**
>
> **Comment:** *“3.The evaluation metric is unfair. By eq(4), the adversarial perturbation will be definitely larger than the baselines. Theorem 1 is obvious and hard to be the contribution of this paper. The magnitude for the baselines should be triple.”*
>
> **Answer:** We would like to thank the reviewer for raising this question. We would first like to clarify two points and then provide experimental evidence.
>
> 1. **All methods are evaluated under the same attacks at test time:** First, we would like to point out that for all our experiments, we use the same adversarial perturbation radius (epsilon) when testing robustness. Therefore, irrespective of how the models were trained, they were all exposed to the same kind of adversarial attacks for comparison.
>
> 2. **A larger perturbation during training does NOT necessarily lead to a more robust model:**
> Although we agree with the need for a fair comparison at test time, we understand that the main criticism comes from the assumption that a larger perturbation during training is likely to induce more robustness. Regarding that, we would like to point out that using larger perturbations does not necessarily translate into a larger robust accuracy at test time, especially with single-step adversarial methods which are prone to catastrophic overfitting.
> We added a new table in Appendix D of our submission where we observe that for all single-step baselines, increasing the perturbation radius does not lead to an increased performance (especially taking into account clean accuracy trade-off). In particular, we observe that when a model catastrophically overfits, it becomes vulnerable to smaller perturbations as well. Thus, increasing the training perturbation budget compared to that of test time would rather lead to catastrophic overfitting earlier (for smaller test epsilon).
>
> 3. **Regarding Theorem 1:** Theorem 1 is not a main contribution of the paper nor was it stated to be so as part of the contributions section in the introduction. It was included in the paper to emphasize our main observation which is that noise does indeed help mitigate catastrophic overfitting, contrary to prior hypothesis which stated that the contribution of noise (combined with clipping) was simply to decrease the magnitude of the perturbations.
>
> **Comment:** *“4. To my understanding, increasing the magnitude of the adversarial perturbation may lead to a worse clean accuracy. Figure 5 verifies the clean accuracy drop. However, the table in Figure 4 seems to illustrate that the clean accuracy of N-FGSM is maintained compared with the baselines. And the table is very hard to understand.”*
>
> **Answer:** We agree that the general empirical observations point towards a trade-off between robustness against larger attacks and clean accuracy - and we do observe this phenomenon in our work as well (for instance, see Figure 5). However, these observations are based on increasing the magnitude of the perturbation induced by the *adversarial attack*. On the other hand, we can think of N-FGSM as first augmenting the sample with noise, and then further augmenting the resulting sample by computing an FGSM attack with step size alpha=epsilon by default. While this leads to a larger perturbation, the entire increased perturbation does not necessarily align with an adversarial direction. We would like to note that perturbing samples with random noise has a significantly milder effect on the clean accuracy than perturbing them with an adversarial attack.
>
> To support our claims, we have included further ablations in the Appendix C of our submission. Where we observe that training with only random noise augmentation does not degrade the adversarial accuracy as much as training with adversarial attacks. We also refer the Reviewer to Figure 12 in the appendix where we observe that moving in random directions along the input space has a much lower impact on the loss than moving along the FGSM direction.
>
>
> **Comment:** *“5. What is the epsilon axis in Figure 3 and 4? Does it represent the magnitude of epsilon for training? If yes, why the robustness decreases as the epsilon increases? If the epsilon is the magnitude of the evaluation, then PGD-50-10 usually represents a PGD attack with 50 steps and 10/255 l-infi norm. It isn't very clear here.”*
>
> **Answer:** Thank you for your question. In all experiments, we use the same value of epsilon for train and test (unless stated otherwise). We have changed the figure labels to make it clear. Regarding PGD-50-10 we adopted the same notation as seen in other works and it means PGD with 50 iterations and 10 restarts. We had defined it in the 7th line of section 4.1, where it first appeared. Now we define it again in the experiments section to avoid confusion.

---

> > ### Author Response · Authors · 2021-11-14
> > **Response 2 of 2**
> >
> > **Comment:** *“1. The contribution of this paper is limited. This paper provides little insight as a research paper as the two improvements of N-FGSM are empirical and heuristic to some extent.”*
> >
> > **Comment:** *“2.The motivation of this paper is unclear. The authors raised the CO problem but turn to the two "key components". The two so-called "key components" of adversarial training, clipping and random steps, are unconvincing. It is more like achieving the improvements first and then finding the motivations to fit them in.”*
> >
> > **Answer:** Although we agree that the main contributions of this paper are empirical, we kindly ask the reviewer to consider:
> >
> > 1. **The empirical novelty in our results.**  Contrary to prior hypotheses, we found a rather surprising result that shows noise is indeed a very simple and useful tool to avoid catastrophic overfitting and we hope can lead to future work to better understand catastrophic overfitting and how noise augmentation fits in adversarial training.
> > 2. **The usefulness of our method:** Adversarial robustness is indeed an important research topic, however it is very expensive. Finding effective methods to conduct adversarial training can boost progress in this field. We are able to match the previous state-of-the-art method (GradAlign) with a 3x speed-up and our method is very easy to implement. Potentially benefiting many researchers working on robustness.
> > 3. **The thorough comparison to related work.** In recent years, several methods have been proposed to perform single-step adversarial training while avoiding catastrophic overfitting. However, not all of them compare on a wide range of perturbation radii or with several networks. We present a large experimental suite to thoroughly compare these baselines under a wide range of settings, spending 18720 GPU hours.
> >
> > **Citations:**
> >
> > (GradAlign) Maksym Andriushchenko and Nicolas Flammarion. Understanding and improving fast adversarial training. In Neural Information Processing Systems (NeurIPS), 2020.

---

### Official Review · Reviewer_ZY47 · 2021-11-02

**Correctness:** 4
**Technical Novelty And Significance:** 3
**Empirical Novelty And Significance:** 3
**Recommendation:** 6
**Confidence:** 5

**Main Review:**

##########################################################################

Pros:

1. The paper attempts to improve the efficiency for adversarial training. For me, the problem itself is real and important.

2. This paper proposes a method called Noise-FGSM (N-FGSM), which attacks noise-augmented samples directly using a single-step. N-FGSM is simple but effective.

3. Extensive experiments shows N-FGSM achieves the SOTA results while achieving a 3x speed-up.

##########################################################################

Cons:

1. Although the authors give theoretical analysis to understand the role of noise in
single-step approaches, the theoretical analysis on that increasing the noise magnitude
and not clipping prevents catastrophic overfitting is not provided. It will be better to explore theoretical justification behind this.

2. Another concern lies in the accuracy on CIFAR 100 and SVHN datasets. As shown in Fig.3, the performance improvement is very slight.


**Summary Of The Paper:**

Different from previous intuitions, this paper find that not clipping the perturbation around the clean sample and using a stronger noise is highly effective in avoiding CO for large perturbation radii. Based on these observations, the authors propose a method called Noise-FGSM (N-FGSM), which achieves the comparable results to GradAlign while achieving a 3x speed-up.

**Summary Of The Review:**

Overall, I vote for accepting. This paper find that not clipping the perturbation around the clean sample and using a stronger noise is highly effective in avoiding CO for large perturbation radii, which is interesting. Based on this, it proposes a simple but effective method called N-FGSM, which achieves the SOTA results while achieving a 3x speed-up. One major concern is about the theoretical analysis on the observations. Hopefully the author can address my concern in the rebuttal period.

---------------------------------------------------------------------------------------------------------------------------------------------------------------------------
UPDATE

I have carafully read your response. My main concern has not been well addressed. The main drawback of this paper is the lack of theoretical analysis. The reason why it works is not clear enough. I believe that the paper may have potential but as its current form has some weakness. I turn to the rating of marginally above the acceptance threshold.

By the way, the performance improvement refers to accuracy rather than efficiency.

---

> ### Author Response · Authors · 2021-11-14
> **Response 1 of 1**
>
> **Comment:** *“Although the authors give theoretical analysis to understand the role of noise in single-step approaches, the theoretical analysis on that increasing the noise magnitude and not clipping prevents catastrophic overfitting is not provided. It will be better to explore theoretical justification behind this.“*
>
> **Answer:** We agree with the Reviewer that it would indeed be a promising direction to explore more thoroughly what exactly is the role of noise in avoiding catastrophic overfitting. Based on our experimental and analytical results we could determine that, contrary to prior intuition, noise can indeed play a fundamental role in avoiding Catastrophic Overfitting. Intuitively, N-FGSM can be viewed as combining noise and adversarial augmentation, where initially the image is augmented with noise and then, to make the perturbation more meaningful (for adversarial training), we further perturb that augmented sample with a weak attack (FGSM).
>
> Previous work [Gilmer et al.] has established a close link between being robust to noise and being robust to adversarial attacks, thus, this combination could potentially lead to better robustness.
> On the other hand, catastrophic overfitting is hypothesized to be an overfitting issue where the model somehow learns a shortcut to be robust to single-step attacks while being vulnerable to multi-step ones. Our intuition here would be that the initial random step is introducing more variability to the perturbations which helps avoid this overfitting. We would like to exhaustively explore these hypotheses but we consider this should be part of future work. Regarding this submission, we kindly ask the reviewer to consider our other contributions such as:
> 1. **The usefulness of our method**: Adversarial robustness is indeed an important research topic, however it is very expensive. Finding effective methods to conduct adversarial training can boost progress in this field. We are able to match previous state-of-the-art method (GradAlign) with a 3x speed-up and our method is very easy to implement. Potentially benefiting many researchers working on robustness.
> 2. **The empirical novelty in our results.** Contrary to prior hypotheses, we found a rather surprising result that shows noise is indeed a very simple and useful tool to avoid catastrophic overfitting and we hope can lead to future work to better understand catastrophic overfitting and how noise augmentation fits in adversarial training.
> 3. **The thorough comparison to related work.** In recent years several methods have been proposed to perform single-step adversarial training while avoiding catastrophic overfitting. However, not all of them compare on a wide range of perturbation radii or with several networks. We present a large experimental suite to thoroughly compare these baselines under a wide range of settings, spending 18720 GPU hours.
>
> **Comment:** *“Another concern lies in the accuracy on CIFAR 100 and SVHN datasets. As shown in Fig.3, the performance improvement is very slight.”*
>
> **Answer:**  As the Reviewer has mentioned in the “Pros:” we “ improve the efficiency for adversarial training.” That is to say, we match the accuracy of state-of-the-art GradAlign while being 3 times faster. This is consistent on all datasets including CIFAR100 and SVHN (Figures 1 and 3). Could we please ask the reviewer to clarify what they meant by “performance improvement”?
>
> **Citations:**
>
> (GradAlign) Maksym Andriushchenko and Nicolas Flammarion. Understanding and improving fast adversarial training. In Neural Information Processing Systems (NeurIPS), 2020.
>
> Justin Gilmer, Nicolas Ford, Nicholas Carlini, and Ekin Cubuk. Adversarial examples are a natural consequence of test error in noise. In International Conference on Machine Learning (ICML), 2019.

---

### Official Review · Reviewer_iWaX · 2021-11-02

**Correctness:** 2
**Technical Novelty And Significance:** 2
**Empirical Novelty And Significance:** 3
**Recommendation:** 5
**Confidence:** 4

**Main Review:**

Strengths:
+ Overall, this paper is well written.
+ Empirical results look fairly good.

Weaknesses:
- The absence of clipping operation will violate the $\ell_p$-ball constraints on perturbation and therefore lead to an unfair comparison with the baselines. The perturbation of adversarial examples is by convention, constrained in an \ell_p norm ball, to indicate the strengths of adversarial attacks and the clipping operation is the key to maintaining the constraint. A larger $\epsilon$ represents larger perturbation budgets and always leads to stronger attacks. Likewise, perturbation without clipping in adversarial training naturally equips models with better robustness. Therefore, although N-FGSM indeed brings better robustness, it may not be fair to directly compare N-FGSM with its baselines. The choice of very large $\eta$ (e.g. 2$\epsilon$) in Algorithm 1, further equipped the adversarial perturbation with a larger attack budget. The real perturbation budget of N-FGSM is likely to be larger than $\epislon$ itself. It will be better for the authors to present the real $\ell_p$ norm of perturbations generated by N-FGSM.

- Based on the above, the role of noise in promoting robustness is also suspicious, because it is hard to tell whether the presence of noise or a larger budget contributes more to better robustness.




**Summary Of The Paper:**

This paper methodically studied the catastrophic overfitting in fast adversarial training (Fast-AT), and revisited the role of noise and clipping operation in Fast-AT. Based on the empirical findings, this paper discovered that the absence of clipping as well as using stronger noise could help avoid catastrophic overfitting. The author further proposed Noise-FGSM, utilizing single-step FGSM and noise-augmented samples to generate adversarial examples for training. Empirical studies showed the superiority of N-FGSM both in terms of performance and speed.


**Summary Of The Review:**

In all, although the method proposed in this paper beats the baselines, it may not be a fair comparison and there are some fundamental errors in the experimental settings.

Post-rebuttal adjustment:

Thanks for the additional clarification. I decide to increase my score by a point due to the authors' effortful response.

I did not increase the score to 6 since I am not fully convinced on why noise augmentation + with adequate strength and without clipping is a principled solution to improve the effectiveness of fast adversarial training. I did not penalize its simplicity (this is fine to me), however, its effectiveness should be further justified either empirically or theoretically, e.g., whether or not the proposed design is applicable to fast adversarial training using TRADES-type loss, or other similar and in-depth studies.

---

> ### Author Response · Authors · 2021-11-14
> **Response 1 of 2**
>
> **Comment:** *“The absence of clipping operation will violate the ℓp-ball constraints on perturbation and therefore lead to an unfair comparison with the baselines. The perturbation of adversarial examples is by convention, constrained in an \ell_p norm ball, to indicate the strengths of adversarial attacks and the clipping operation is the key to maintaining the constraint. A larger ϵ represents larger perturbation budgets and always leads to stronger attacks. Likewise, perturbation without clipping in adversarial training naturally equips models with better robustness. Therefore, although N-FGSM indeed brings better robustness, it may not be fair to directly compare N-FGSM with its baselines. The choice of very large η (e.g. 2ϵ) in Algorithm 1, further equipped the adversarial perturbation with a larger attack budget. The real perturbation budget of N-FGSM is likely to be larger than \epislon itself. It will be better for the authors to present the real ℓp norm of perturbations generated by N-FGSM.”*
>
> **Answer:** We would like to thank the Reviewer for raising this question, we will break up the answer to address several points:
>
> 1. **Satisfying the l_p constraint during training is a convention rather than a requirement.** The objective when performing adversarial training is to obtain models that will be adversarially robust within a predefined perturbation set at test time, which is usually defined with an l_p-ball. However, we do not consider the formulation of adversarial training introduced in [Madry et al.] and depicted in Eq(1) to be a hard requirement that all adversarial training methods should follow. In fact, we understand this as a convention or common practice with the rationale that it is not necessary to train the model with perturbations outside of the l_p -ball because they will not be present at test time.  Although this is true at a theoretical level where one solves the inner maximization exactly, when performing single-step adversarial training we are making a rough approximation of the inner maximization. Thus, the previous assumption does not necessarily hold and it might be worth exploring methods that use larger perturbations during training.
>
> 2. **All methods are evaluated under the same attacks at test time:** For all experiments, we use the same epsilon during training and testing. So in that regard, given an epsilon, robustness evaluations are comparable in the sense that all models have been exposed to the same kind of attacks at test time - regardless of how they were trained. In regards to fairness of experiments, it does not matter how models were trained, what matters for a fair comparison is that all models are tested against the same attack strength, i.e. epsilon test.
>
>
> 3. **A larger perturbation during training does NOT necessarily lead to a more robust model:**
> Although intuitively a larger epsilon will lead to stronger attacks, we would like to point out that this does not necessarily translate into a larger robust accuracy at test time, especially with single-step adversarial methods - which are prone to catastrophic overfitting.  While N-FGSM increases the perturbation size, the perturbation is not only adversarial but it also includes noise. The  combination of both is what we find to achieve the reported improved robust accuracies avoiding catastrophic overfitting.
>
>    We added a new table in Appendix D of our submission, where we observe that for all single-step baselines, increasing the perturbation radius does not lead to an increased performance (especially taking into account clean accuracy trade-off). In particular, we observe that when a model catastrophically overfits, it becomes vulnerable to smaller perturbations as well. Thus, increasing the training perturbation budget compared to that of test time would rather lead to catastrophic overfitting earlier (for smaller test epsilon).
>
> 4. **Reporting the actual magnitude of N-FGSM perturbations:** This indeed was theoretically analyzed in Theorem 1 in the main paper with an elaborated discussion in Appendix F showing that N-FGSM enjoys larger perturbations than FGSM and RS-FGSM on average.
>
>
> **Citations:**
>
> Aleksander Madry, Aleksandar Makelov, Ludwig Schmidt, Dimitris Tsipras, and Adrian Vladu.
> Towards deep learning models resistant to adversarial attacks. In International Conference on
> Learning Representations (ICLR), 2018.

---

> > ### Author Response · Authors · 2021-11-14
> > **Response 2 of 2**
> >
> > **Comment:** *“Based on the above, the role of noise in promoting robustness is also suspicious, because it is hard to tell whether the presence of noise or a larger budget contributes more to better robustness.“*
> >
> > **Answer:** Thanks for raising this question. In the previous point, we mentioned that none of the single-step methods benefits from an increase in the perturbation budget during training. Moreover, in Appendix C we present a new table in which we ablate the effect of increasing the FGSM step size vs the noise level in N-FGSM. The experimental ablations suggest that simply increasing the perturbation size does not avoid catastrophic overfitting, on the contrary, increasing the FGSM step size (alpha) is what leads to catastrophic overfitting, which can only be mitigated when coupled by an increase in the noise level (k).

---

> > > ### Comment · Reviewer_iWaX · 2021-11-24
> > > **Thanks for the response**
> > >
> > > Some of my concerns have been properly addressed, however, one of the main concerns remains.
> > >
> > > Yes, I understood that "All methods are evaluated under the same attacks at test time", however, the train-time attack strength is different when comparing the proposal (no l_p projection) with the baselines. Let us take the argument "Satisfying the l_p constraint during training is a convention rather than a requirement", does this mean that removing projection should be a **principled** solution for many robust training methods, e.g., TRADES https://arxiv.org/abs/1901.08573? And should the baselines like GradAlign also use a similar train-time attack setup for comparison?
> > >
> > > Thanks,

---

> > > > ### Author Response · Authors · 2021-11-24
> > > > **Authors reply**
> > > >
> > > > **Question:** *"Should removing the projection step be a principled solution for many robust training methods?"*
> > > >
> > > > **Answer:**  In general, we do not find that removing the projection step can be used as a principled solution for **all** methods. For instance, **removing clipping from FGSM and RS-FGSM can not prevent catastrophic overfitting** (blue and orange plots in Figure 2 right, respectively). This is also clear from the first column in Table 1 in Appendix C; note that increasing the perturbation size $\alpha$ for FGSM under no clipping also results in catastrophic overfitting.
> > > >
> > > > We would like to emphasize that our method does not merely remove the projection step but it also increases the perturbation size with **noise** -- not in an adversarial direction -- which is key to the success of our method.
> > > >
> > > > **Question:** *"Should the baselines like GradAlign also use a similar train-time attack setup for comparison?"*
> > > >
> > > > **Answer:** Following the reviewers initial comments we added a new table (Table 2) in Appendix D to address this very question. We observe that **increasing the $\epsilon_{\text{train}}$ with respect to the $\epsilon_{\text{test}}$ yields worse results** for all single-step baselines, including GradAlign. Therefore, the answer to the question is negative. For more details refer to point 3 of our previous answer and Appendix D.

---

> > > > > ### Comment · Reviewer_iWaX · 2021-11-26
> > > > > **Thanks.**
> > > > >
> > > > > Thanks for the response.
> > > > >
> > > > > "We would like to emphasize that our method does not merely remove the projection step but it also increases the perturbation size with noise -- not in an adversarial direction -- which is key to the success of our method."
> > > > >
> > > > > May I understand that the proposed modifications to fast adversarial training are so far not applicable to the other single-step baselines? Can authors make some more in-depth explanations about this specialty?

---

> > > > > > ### Author Response · Authors · 2021-11-29
> > > > > > **Authors reply**
> > > > > >
> > > > > > **Question:** *"May I understand that the proposed modifications to fast adversarial training are so far not applicable to the other single-step baselines? Can authors make some more in-depth explanations about this specialty?"*
> > > > > >
> > > > > > **Answer:** To clarify, we indeed can apply our modifications to other single-step approaches. However, since other single-step approaches are variants of the well known FGSM [Goodfellow et al.], applying such modifications to existing single-step baselines will simply recover N-FGSM (or vice-versa). For example:
> > > > > >
> > > > > >  - Setting $k=0$ (no noise) and step size $\alpha=\epsilon$ in our method (N-FGSM), we would recover FGSM.
> > > > > >
> > > > > >  - On the other hand, if N-FGSM sets the noise level $k = \epsilon$ and adds a clipping step to project the perturbation onto the $l_{\infty}$ ball then it would recover RS-FGSM [Wong et al.].
> > > > > >
> > > > > >  - Even if we apply our modifications to GradAlign [Andriushchenko and Flammarion] which uses FGSM with an expensive regularizer, it’s computational cost will still be 3x that of FGSM, therefore, will not give us the desired efficiency we are looking for.
> > > > > >
> > > > > > We would like to reiterate the strengths of our contributions for better clarity:
> > > > > >
> > > > > >  - Though our modifications might seem simple, they are highly effective. With our modifications, we can obtain an algorithm for adversarial training that is single-step and is as effective as the SOTA GradAlign which requires 3x training time.
> > > > > >
> > > > > >  - Our observations that noise augmentation can help avoid catastrophic overfitting if applied properly (with an adequate strength and without clipping) were not obvious. Note that previous state of the art (GradAlign) had concluded that the role of noise was merely to reduce the magnitude of the perturbations during training.
> > > > > >
> > > > > >  Our suggested modifications, though simple, were largely overlooked by the community. We provide insights and modifications that no prior art showed before, and these modifications do improve the performance as we have shown via a variety of experiments.
> > > > > >
> > > > > >
> > > > > >
> > > > > > **Citations:**
> > > > > >
> > > > > > Eric Wong, Leslie Rice, and J. Zico Kolter. Fast is better than free: Revisiting adversarial training. In International Conference on Learning Representations (ICLR), 2020
> > > > > >
> > > > > > Ian Goodfellow, Jonathon Shlens, and Christian Szegedy. Explaining and harnessing adversarial examples. International Conference on Learning Representations (ICLR), 2015.
> > > > > >
> > > > > > Maksym Andriushchenko and Nicolas Flammarion. Understanding and improving fast adversarial training. In Neural Information Processing Systems (NeurIPS), 2020.

---

> > > > > > > ### Comment · Reviewer_iWaX · 2021-11-29
> > > > > > > **Thanks for the follow-up response**
> > > > > > >
> > > > > > > Thanks for the additional clarification. I decide to increase my score by a point due to the authors' effortful response.
> > > > > > >
> > > > > > > I did not increase the score to 6 since I am not fully convinced on why noise augmentation + with adequate strength and without clipping is a principled solution to improve the effectiveness of fast adversarial training. I did not penalize its simplicity (this is fine to me), however, its effectiveness should be further justified either empirically or theoretically, e.g., whether or not the proposed design is applicable to fast adversarial training using TRADES-type loss https://arxiv.org/pdf/2010.01278.pdf , or other similar and in-depth studies.

---

### Official Review · Reviewer_6yud · 2021-11-03

**Correctness:** 3
**Technical Novelty And Significance:** 3
**Empirical Novelty And Significance:** 2
**Recommendation:** 5
**Confidence:** 2

**Main Review:**

The paper is overall well-written and easy to follow. The proposed methodology is quite simple and easy to implement. Detailed and comprehensive empirical evidence is provided to support the proposed method. However, I have several major comments regarding the presentation and intuition.


* From my understanding, the proposed method can be rephrased as using a training perturbation radius significantly larger than the test perturbation radius (typically twice as mentioned by the authors), since the perturbation is randomly initialized in a larger radius, and no clipping is used. Could the authors confirm if my understanding is correct?


* The paper found that a larger training perturbation radius can mitigate the catastrophic overfitting. This is somewhat counter-intuitive. I was expecting a figure clearly showing that as the training perturbation becomes larger, the catastrophic overfitting is mitigated (probably redesign Figure 2 right). I think more (empirical) analyses will be helpful to understand why this is true. There are some analyses in Section 4.1 on the loss landscape, but the intuition is still lacking.


* I don't quite understand why a larger training perturbation radius will not lead to a clean accuracy drop. I think this observation is quite important to the paper's contribution but there is almost no justification. Providing at least some empirical analyses would be helpful.


Minor comments:
* The observation that larger perturbation does not necessarily hurt the performance is important in this paper. Section 4.2 follows [1] and show such observation is contradictory to their claim. It is better to merge this part of the analysis with Section 4.1 and provide a more integrated and intuitive understanding.


* The notation epsilon is abused without clearly stating whether it is training perturbation or test perturbation. In Figure 2 I believe the epsilon here refers to only the test perturbation radius while in Figure 3 it could be both training and test?


* Consider include some numerical results in the main paper. For example, move part of the performance tables in Appendix J to the experiment section, along with the training cost or speed up factor. This would justify the proposed method better if the main claim is efficiency.










**Summary Of The Paper:**

This paper aims to address a failure mode in the traditional single-step adversarial training known as catastrophic overfitting. They show that compared to the common practice of generating the adversarial perturbation, adopting larger random initialization and avoiding clipping the perturbation can effectively mitigate catastrophic overfitting. The effects of these two techniques are analyzed empirically in detail, followed by a comprehensive comparison to other methods.

**Summary Of The Review:**

I believe this paper made some interesting observations towards understanding and mitigating catastrophic overfitting in adversarial training. However, I am not fully convinced as the support analyses cannot provide sufficient evidence of why the proposed method could work. I am willing to increase my score if the authors can better clarify the proposed method and provide more intuition.

---

> ### Author Response · Authors · 2021-11-14
> **Response 1 of 3**
>
> **Comment**: *“From my understanding, the proposed method can be rephrased as using a training perturbation radius significantly larger than the test perturbation radius (typically twice as mentioned by the authors), since the perturbation is randomly initialized in a larger radius, and no clipping is used. Could the authors confirm if my understanding is correct?”*
>
> **Answer**: While our method (N-FGSM) indeed leads to larger perturbations compared to other baselines, it would be an oversimplification to simply rephrase it as “using a training perturbation radius significantly larger than the test perturbation one”. This is due to the fact that the perturbations of N-FGSM are larger **due to noise**, which will not generally follow an adversarial direction. Thus, N-FGSM is not necessarily equivalent to augmenting with other adversarial attacks with a similarly larger perturbation size. Below we provide details of additional experiments to support this.
>
> In Appendix C (newly added) from our submission, we present a new table where we ablate the effect of increasing the FGSM step size vs the noise level in N-FGSM. These results show that increasing the perturbation size in FGSM does not avoid catastrophic overfitting; in fact, increasing the FGSM step size is the factor leading to catastrophic overfitting in the first place, which can only be mitigated when coupled by an increase in the noise level.
>
> **Comment**: *“The paper found that a larger training perturbation radius can mitigate the catastrophic overfitting. This is somewhat counter-intuitive. I was expecting a figure clearly showing that as the training perturbation becomes larger, the catastrophic overfitting is mitigated (probably redesign Figure 2 right). I think more (empirical) analyses will be helpful to understand why this is true. There are some analyses in Section 4.1 on the loss landscape, but the intuition is still lacking.”*
>
> **Answer**: We would like to emphasize that we did not merely find that increasing the perturbation radius led to a mitigation of catastrophic overfitting (please see our answer to the first point). What we observe is that we need to do both i) remove clipping **AND** ii)  increase the noise level to avoid catastrophic overfitting. Looking at Figure 2, we can observe that:
>
> 1. **Increasing noise is not enough**: If we increase the noise level without modifying the clipping, we observe (Fig 2 Left) that the final robustness on the test set decreases. For example, Fig 2 (left) shows that for a clipping radius of 1 epsilon, increasing the noise leads to worse performance.
>
> 2. **Removing clipping is not enough**: In Fig 2 (left) we also observe that, for any noise level, increasing the radius of clipping (up to the limit of “no clipping”) has a clear positive impact on performance. However, in Fig 2 (right), we show that removing clipping without increasing the noise level (Noise of 1 epsilon) can not avoid catastrophic overfitting either.

---

> > ### Author Response · Authors · 2021-11-14
> > **Response 2 of 3**
> >
> > **Comment**:  *“I think more (empirical) analyses will be helpful to understand why this is true. There are some analyses in Section 4.1 on the loss landscape, but the intuition is still lacking”*
> >
> > **Answer**: Previous intuition regarding the role of noise in avoiding CO was that it simply decreased the perturbation radius  [Andriushchenko & Flamarion].  Based on our experimental and analytical results we could determine that noise does indeed play a fundamental role in avoiding Catastrophic Overfitting, despite leading to larger perturbations. Thus, we do agree with the Reviewer that the results are surprising. Intuitively N-FGSM can be viewed as a combination of noise injection and adversarial augmentation, where initially the image is injected with noise and then, to make the perturbation more meaningful (for the sake of adversarial training), we further augment the noisy sample with a single-step attack (FGSM).
> >
> > Previous work [Gilmer et al.] has established a close link between robustness to noise and robustness to adversarial attacks, thus, this combination could potentially lead to better robustness.
> > On the other hand, catastrophic overfitting seems to be indeed an overfitting issue where the model somehow learns a shortcut to be robust to single-step attacks while being vulnerable to multi-step ones. Our intuition here would be that the initial random step is introducing more variability to the perturbations which helps avoid such overfitting. We are interested in exhaustively exploring these hypotheses for future work. Regarding this submission, we believe that our empirical contributions stand alone, and we recap them as:
> >
> > 1. **The empirical novelty in our results**.  Contrary to prior hypotheses, we found a rather surprising result that shows noise is indeed a very simple and useful tool to avoid catastrophic overfitting and we hope can lead to future work to better understand catastrophic overfitting and how noise augmentation fits in adversarial training.
> >
> > 2. **The usefulness of our method**: Adversarial robustness is indeed an important research topic, however it is very expensive. Finding effective methods to conduct adversarial training can boost progress in this field. We are able to match the previous state-of-the-art method (GradAlign) with a 3x speed-up and our method is very easy to implement. Potentially benefiting many researchers working on robustness.
> >
> > 3. **The thorough comparison to related work**. In recent years, several methods have been proposed to perform single-step adversarial training while avoiding catastrophic overfitting. However, not all of them compare on a wide range of perturbation radii or with several networks. We present a large experimental suite to thoroughly compare these baselines under a wide range of settings, spending 18720 GPU hours.
> >
> > **Comment**: *“I don't quite understand why a larger training perturbation radius will not lead to a clean accuracy drop. I think this observation is quite important to the paper's contribution but there is almost no justification. Providing at least some empirical analyses would be helpful.”*
> >
> > **Answer**: We agree that the general empirical observations point towards a trade-off between robustness against larger attacks and clean accuracy - and we do observe this phenomenon in our work as well (for instance, see Figure 5). However, these observations are based on increasing the magnitude of the perturbation induced by the *adversarial attack*. On the other hand, we can think of N-FGSM as first augmenting the sample with noise, and then further augmenting the resulting sample by computing an FGSM attack with step size alpha=epsilon by default. While this leads to a larger perturbation, the entire increased perturbation does not necessarily align with an adversarial direction. We would like to note that perturbing samples with random noise has a significantly milder effect on the clean accuracy than perturbing them with an adversarial attack.
> >
> > To support our claims, we have included further ablations in the Appendix C of our submission. Where we observe that training with only random noise augmentation does not degrade the adversarial accuracy as much as training with adversarial attacks. We also refer the Reviewer to Figure 12 in the appendix where we observe that moving in random directions along the input space has a much lower impact on the loss than moving along the FGSM direction.
> >
> > **Citations:**
> >
> > Maksym Andriushchenko and Nicolas Flammarion. Understanding and improving fast adversarial training. In Neural Information Processing Systems (NeurIPS), 2020.
> >
> > Justin Gilmer, Nicolas Ford, Nicholas Carlini, and Ekin Cubuk. Adversarial examples are a natural consequence of test error in noise. In International Conference on Machine Learning (ICML), 2019.

---

> > > ### Author Response · Authors · 2021-11-14
> > > **Response 3 of 3**
> > >
> > > **Comment**: *“The observation that larger perturbation does not necessarily hurt the performance is important in this paper. Section 4.2 follows [1] and shows such observation is contradictory to their claim. It is better to merge this part of the analysis with Section 4.1 and provide a more integrated and intuitive understanding.”*
> > >
> > > **Answer**: Thank you for the suggestion, we have added experiments and discussion in Appendix C in the paper and referred them in the main text so that readers can look for more details.
> > >
> > > **Comment**: *“The notation epsilon is abused without clearly stating whether it is training perturbation or test perturbation. In Figure 2 I believe the epsilon here refers to only the test perturbation radius while in Figure 3 it could be both training and test?”*
> > >
> > > **Answer**: Thank you for pointing this out, in both figures we use the same epsilon at train and test time. We have changed the figure labels to make them clearer.
> > >
> > > **Comment**: *“Consider include some numerical results in the main paper. For example, move part of the performance tables in Appendix J to the experiment section, along with the training cost or speed up factor. This would justify the proposed method better if the main claim is efficiency.”*
> > >
> > > **Answer**: Thank you for the suggestion, we kindly refer the reviewer to Figure 1 (right) where we report the relative cost of our method and different baselines.

---

> > > > ### Comment · Reviewer_6yud · 2021-11-30
> > > > **Thank you for the detailed responses**
> > > >
> > > > I think the newly added two tables in the appendix provide much more information to evaluate the contribution of this paper.
> > > >
> > > > Table 1 clearly shows that increasing the noise level helps mitigate catastrophic overfitting. On the other hand, increasing the step size, which is equivalent to increasing the perturbation size since no clipping is used, benefits the robust accuracy but facilitates catastrophic overfitting. However, I think these two observations are already presented in the literature. The fact that random noise helps mitigate catastrophic overfitting is carefully studied in [1]. And increasing the perturbation size benefits the robust accuracy is a widely recognized phenomenon, to the best of my knowledge.
> > > >
> > > > Table 2 compares the proposed method (N-FGSM) with other fast adversarial training methods. I am particularly interested in the comparison between N-FGSM and RS-FGSM [1] since N-FGSM can be viewed as a variant of RS-FGSM and they have similar computational efficiency. However, in terms of the performance, N-FGSM doesn't show a significant advantage over RS-FGSM. It is more like trading the clean accuracy for robust accuracy since N-FGSM removes the clipping step. In terms of mitigating catastrophic overfitting, I believe more experiments on the baseline methods should be provided. In particular, will increasing the noise level in RS-FGSM also mitigate the catastrophic overfitting?
> > > >
> > > > Overall, given its empirical nature, I believe the contributions made in this paper might not be novel. Therefore I will keep my score as it is. I would like to see more clear ablation studies as in Table 1 and Table 2 to demonstrate the contributions of this work compared to the literature.
> > > >
> > > > [1] Fast Is Better Than Free: Revisiting Adversarial Training. Wong et al., 2020.

---

> > > > > ### Author Response · Authors · 2021-11-30
> > > > > **Authors' response**
> > > > >
> > > > > **Answer:** We appreciate that the reviewer found the tables useful, however we would like to respectfully disagree with some of the conclusions:
> > > > >
> > > > > **Regarding Table 1**:
> > > > >
> > > > >  - The role of noise has been indeed previously studied in the literature [1, 2], however perturbations were always constrained to the $l_\infty$ ball by construction or via clipping. The results presented in table 1 do not have clipping and explore larger noise levels. **While this is something indeed simple, it was not obvious and is highly effective.** Note that previous state of the art [3] had concluded that the role of noise was merely to reduce the magnitude of the perturbations during training, which we show is not the case.
> > > > >
> > > > >  - The initial observation that noise can be helpful to mitigate catastrophic overfitting is presented in [1] however, if the reviewer goes to their appendix A figure 3 they will note how **when increasing the step size with RS-FGSM leads to catastrophic overfitting for large step sizes** and they do not provide any hints as to whether some adjustment to noise or clipping can avoid that. On the other hand, on table 1 **we observe with an adequate noise level and without clipping we can avoid catastrophic overfitting.**
> > > > >
> > > > >  **Regarding Table 2**:
> > > > >
> > > > >  -  The objective of Table 2 is mainly to ablate all single-step methods when using a training epsilon which was larger than the test epsilon and the conclusion is that in general it can not avoid catastrophic overfitting. In particular, note that RS-FGSM obtains 0% robust accuracy on the second and third columns.
> > > > >
> > > > >  - **“Will increasing the noise level in RS-FGSM also mitigate the catastrophic overfitting?”** Please refer to Figure 2 (Left) where we increase the noise level along with the clipping radius. Note that when clipping radius is set to $1\epsilon$ this would be equivalent to varying the noise level with RS-FGSM and we observe that robust accuracy is significantly affected.
> > > > >
> > > > >  - Regarding more experiments on the baseline methods, we would like to emphasize that all  other single-step approaches are variants of the well known FGSM [4], therefore they could be modified with more noise and without clipping as we suggest in N-FGSM, but then we would recover N-FGSM.
> > > > >
> > > > > We would like to reiterate the strengths of our contributions for better clarity:
> > > > >
> > > > >  - Though our modifications might seem simple, they are highly effective. With our modifications, we can obtain an algorithm for adversarial training that is single-step and is as effective as the SOTA GradAlign which requires 3x training time.
> > > > >
> > > > >  - Our observations that noise augmentation can help avoid catastrophic overfitting( if applied properly with an adequate strength and without clipping) were not obvious. Note that previous state of the art (GradAlign) had concluded that the role of noise was merely to reduce the magnitude of the perturbations during training.
> > > > >
> > > > > Our suggested modifications, though simple, were largely overlooked by the community. We provide insights and modifications that no prior art showed before, and these modifications do improve the performance as we have shown via a variety of experiments.
> > > > >
> > > > >
> > > > > **Citations**
> > > > >
> > > > > [1] Fast Is Better Than Free: Revisiting Adversarial Training. Wong et al., 2020.
> > > > >
> > > > > [2] Florian Tramer, Alexey Kurakin, Nicolas Papernot, Ian Goodfellow, Dan Boneh, and Patrick Mc- ` Daniel. Ensemble adversarial training: Attacks and defenses. In International Conference on Learning Representations (ICLR), 2018.
> > > > >
> > > > > [3] Maksym Andriushchenko and Nicolas Flammarion. Understanding and improving fast adversarial training. In Neural Information Processing Systems (NeurIPS), 2020.
> > > > >
> > > > > [4] Ian Goodfellow, Jonathon Shlens, and Christian Szegedy. Explaining and harnessing adversarial examples. International Conference on Learning Representations (ICLR), 2015.

---

### Author Response · Authors · 2021-11-14
**Summary of changes**

We would like to thank all the Reviewers for their time and comments. To address the Reviewers points we have made several updates on the paper:
 - All relevant figures now explicitly state that the same epsilon is used for training and evaluation in the x axis label to avoid confusion.

 - We define the PGD-50-10 attack again at the start of the experiment section for clarity.

 - We have added two new sections in the appendix (Appendix C and Appendix D) with further experiments that address reviewers concerns. New text is marked in red for easier identification.

**Appendix C**: We present a table where we ablate the effect of increasing the FGSM step size vs the noise level in N-FGSM. These results show that increasing the perturbation size in FGSM *does not avoid catastrophic overfitting*; in fact, increasing the FGSM step size is the factor leading to catastrophic overfitting in the first place, which can only be mitigated when coupled by an increase in the noise level. Moreover, we observe how training with noise injected images has a much milder effect on clean accuracy than training with adversarially perturbed images.

**Appendix D**: We present a table reporting the clean and robust accuracy for all single-step methods as we increase the training budget. We observe that, considering the trade-off between clean and robust accuracy, all methods perform best when training with the same epsilon to be applied at test time. In particular, we observe that when a model catastrophically overfits, it becomes vulnerable to smaller perturbations as well. Thus, increasing the training perturbation budget compared to that of test time would rather lead to catastrophic overfitting earlier (for smaller test epsilon).

---

### Decision · Program_Chairs · 2022-01-20

**Decision:**

Reject

**Comment:**

This paper aims to address the catastrophic overfitting issue in single-step adversarial training. Specifically, this paper finds that 1) using larger random noise initialization and 2) avoiding clipping adversarial perturbations are the two keys for stabilizing single-step adversarial training.

Overall, the reviewers find this paper is well-written and the empirical results look promising. The reviewers originally misunderstood certain technical details of this paper, but got clarified in the discussion period.  However, the biggest concern shared by the reviewers is that the motivation of using larger random noise initialization and avoiding clipping adversarial perturbation is pretty unclear---they all fail to (either empirically or theoretically) understand how and why these two techniques are helpful to preventing catastrophic overfitting. Given the main contribution of this paper is a revisiting of existing techniques, it is a legitimate concern from the reviewer side for demanding the in-depth empirical analysis or the theoretical proof to help them better understand the proposed method; otherwise, the novelty contribution of this paper may get trivialized.

I encourage the authors to delve deeper into the proposed method and make a stronger submission next time.